# Sliding of temperate basal ice on a rough, hard bed: creep mechanisms, pressure melting, and implications for ice streaming

Maarten Krabbendam

British Geological Survey, Lyell Centre, Research Avenue South, Edinburgh EH14 4AP, UK

*Correspondence to*: Maarten Krabbendam (mkrab@bgs.ac.uk)

**Abstract**

Basal ice motion is crucial to ice dynamics of ice sheets. The classic Weertman model for basal sliding over bedrock obstacles proposes that sliding velocity is controlled by pressure melting and/or ductile flow, whichever is the fastest; it further assumes that pressure melting is limited by heat flow through the obstacle and ductile flow is controlled by standard power-law creep. These last two assumptions, however, are not applicable if a substantial basal layer of temperate ($T \sim T_{melt}$) ice is present. In that case, frictional melting can produce excess basal meltwater and efficient water flow, leading to near-thermal equilibrium. High-temperature ice creep experiments have shown a sharp weakening of a factor 5-10 close to $T_{melt}$, suggesting standard power-law creep does not operate due to a switch to melt-assisted creep with a possible component of grain boundary melting. Pressure melting is controlled by melt water production, heat advection by flowing meltwater to the next obstacle, and heat conduction through ice/rock over half the obstacle height. No heat flow through the obstacle is required. Ice streaming over a rough, hard bed, as possibly in the Northeast Greenland Ice Stream, may be explained by enhanced basal motion in a thick temperate ice layer.

## 1 Introduction

The manner in which ice deforms within an ice sheet and moves or slides over its base are critical to accurately model the dynamic past, present and future behaviour of such ice bodies (e.g., Marshall, 2005). Internal deformation of cold ice is fairly well understood, and predictions made on the basis of physical laws (e.g., 'Glen's flow law') are broadly confirmed by observations (e.g., Dahl-Jensen and Gundestrup, 1987; Paterson, 1994; Cuffey and Paterson, 2010; Ryser et al., 2014, but see Paterson (1991) for problems with dusty ice, and Hooke (1981) for a general critique). This is not the case for basal sliding,

for which many parameters are poorly constrained (e.g. Cuffey and Paterson, 2010). Instead, many models of modern ice sheets use an empirical drag factor or slip coefficient, derived from observed ice velocity and estimated shear stresses (e.g., MacAyeal et al., 1995; Gudmundsson and Raymond, 2008; Ryser et al., 2014). Using an empirical drag factor is reasonable to describe and understand present-day, near-instantaneous ice sheet behaviour, but cannot reliably predict or reconstruct ice
velocities if parameters such as ice thickness, driving forces and meltwater production change significantly.

This problem is particularly acute for ice streams with poor topographic steering. For such ice streams it is commonly assumed that the necessary low drag at their base can be explained by the presence of soft sediment or deformable till (e.g., Alley et al., 1987; Hindmarsh, 1997; Winsborow et al., 2010), which has indeed been shown to occur below some ice
streams in West Antarctica (e.g. Alley et al., 1987; King et al., 2009) and also documented in the geomorphological record (e.g. Margold et al., 2015). However, there is increasing geomorphological evidence for palaeo-ice streaming without clear topographic steering on rough, hard bedrock-dominated beds. Hard, rough beds are widespread on the beds of the former Pleistocene ice sheets and also likely beneath large parts of the present-day Greenland and Antarctic ice sheets (e.g., Kleman et al., 2008; Eyles, 2012; Rippin, 2013; Krabbendam and Bradwell, 2014; Krabbendam et al., 2016). Evidence for palaeo-
ice streaming on hard beds has been reported from the former Pleistocene Laurentide and British ice sheets and deglaciated parts of West Greenland (Smith, 1948; Stokes and Clark, 2003; Roberts and Long, 2005; Bradwell et al., 2008; Eyles, 2012; Bradwell, 2013; Eyles and Putkinen, 2014; Krabbendam et al., 2016). In these areas, the deforming-bed models cannot apply because little or no soft-sediment is present. These palaeo-ice stream zones are surrounded by areas also subjected to less intense, but still warm-based, ice erosion suggesting intermediate ice velocities (e.g., Bradwell, 2013), consistent with
ice velocity analysis and borehole observations from the Greenland Ice Sheet that show significant warm-based sliding (10-100 m yr$^{-1}$) outside ice-streams (Lüthi et al., 2002; Ryser et al., 2014; Joughin et al., 2010). Thus, fast ice flow appears to be possible on hard, rough beds and cannot be explained by a simple cold/warm thermal boundary (cf. Payne and Dongelmans, 1997). In Greenland, the massive Northeast Greenland Ice Stream remains difficult to explain, as current explanations invoke geologically unreasonably high geothermal heat flows (e.g., Fahnenstock et al., 2001) and a deformable bed with an
unknown and questionable till source (Christianson et al., 2014).

A solution may be presented by the presence of a basal layer of temperate ice (ice at the melting temperature $T_m$), below cold ice that makes up the remainder of the ice sheet. Drilling in Greenland Ice Sheet adjacent to the Jakobshavn Isbrae has recorded a basal layer of temperate ice below cold ice of some 30 m thickness (Lüthi et al., 2002), equal or greater in height
than typical bedrock obstacles (roches moutonnées, whalebacks) in most crystalline gneiss terrains (Krabbendam and Bradwell, 2014). Temperate ice has also been modelled to occur beneath other parts of the Greenland Ice Sheet (e.g., Dahl-Jensen, 1989; Calov and Hutter, 1996; Greve, 1997). Two pertinent questions follow from these observations:

1) How does such a temperate layer develop and how is it maintained, given that it is overlain by cold ice? In-situ measurements at a glacier base and experiments have recently shown that warm-based basal sliding occurs under significant friction, caused by basal-debris / bedrock contacts (Iverson et al., 2003; Cohen et al., 2005; Zoet et al., 2013), generating significant frictional heat at the base, which is important for the development of a temperate ice layer.

2) If basal sliding operates in temperate ice, is the rheology of temperate ice adequately described by a standard power law? How do horizontal thermal gradients, necessary for regelation to proceed, build up? The essence of the classic Weertman (1957) sliding model is that basal ice movement past an obstacle occurs by pressure melting around the obstacle and by ductile flow enhanced by stress concentrations near the obstacle, with ductile flow being more important for larger obstacles. The rheology of enhanced ductile flow is normally described by a standard power law ('Glen's flow law'), whereas pressure melting is regarded to be limited by heat flow *through the obstacle*. Sparse experimental evidence, however, suggests that temperate ice is considerably weaker than cold ice, and that creep may not be modelled reliably according to a standard power law (e.g. Colbeck and Evans, 1973; Duval, 1977; Morgan, 1991). Furthermore, in a basal temperate ice layer it maybe that no thermal gradient across an obstacle can be maintained, and that pressure melting at the stoss side has different thermal controls than in the classic model.

Numerous studies have improved upon the Weertman sliding model, focussing on more realistic geometries of the bedrock obstacles, more sophisticated analyses of the stresses near the obstacles, and models that allow for lee-side cavitation, which was not allowed in the original model (e.g., Kamb and La Chapelle, 1964; Kamb, 1970; Nye, 1970, Lliboutry, 1993; Schoof, 2005). Nevertheless, the mechanical behaviour of temperate ice moving over a rough hard bed remains poorly constrained. This paper deals with four issues. Firstly, the problem of how a temperate layer can grow and be maintained below cold ice is discussed. Secondly, it will be shown that the two critical assumptions of classic Weertman sliding (enhanced ductile flow controlled by a standard power law and pressure melting controlled by heat flow through an obstacle) cannot be applied to temperate ice, and alternative controlling mechanisms are proposed. Thirdly, the implications of the different behaviour of temperate ice below cold ice for thermo-mechanical modelling of ice sheets are discussed. Finally, it is suggested that the development of a temperate basal ice layer may help to explain the occurrence of ice streaming on rough, hard beds, such as seen on deglaciated terrains and possibly also below the Northeast Greenland Ice Stream.

The paper takes a conceptual approach, focussing on the primary thermodynamic and rheological controls, so to achieve an improved conceptual model of basal ice motion around bedrock obstacles, rather than the exact quantification of geometries and stress distributions around such obstacles.

## 2 Basal meltwater production by frictional sliding

Consider sliding over a flat area without obstacles, over which debris-laden ice slides under friction. Friction produces drag and heat. Frictional heat production $Q_{fr}$ and geothermal heat flow $Q_{geo}$ together control the heat budget at the base of ice sheets. Frictional heat production is controlled by the friction coefficient $\mu$, the normal vertical stress $\sigma_{nv}$ and the sliding velocity $V_{sl}$ according to:

(1) $\qquad Q_{fr} = \mu\, \sigma_v\, V_{sl} \approx \mu(Pi - Pw)\, V_{sl}$ in [W m$^{-2}$].

The normal vertical stress $\sigma_v$ can be taken as the effective pressure, that is ice pressure $P_i$ minus water pressure $P_w$. This is the standard friction model, using empirically derived bulk friction coefficients averaged over an area containing many debris particles (e.g. Budd et al., 1979; Cohen et al., 2005; Emerson and Rempel, 2007; Zoet et al., 2014), rather than the theoretical models that consider individual clasts (e.g. Boulton, 1974; Hallet, 1979). Iverson et al. (2003) measured $\mu = 0.05$ *in situ* below a glacier; Budd et al. (1979) measured $\mu = 0.04 - 0.1$ for experimental sliding over rocks of different micro-roughness and Zoet et al. (2013) reported $\mu = 0.01 - 0.05$ for experimental sliding at the pressure melting point, but much higher values ($\mu = 0.2 - 0.6$) at colder temperatures and also, intriguingly, for warm-based sliding of ice over sandstone, which suggest that different bedrock lithologies can result in very different friction coefficients and hence basal melting rates. Emerson and Rempel (2007) observed high friction coefficients of $\mu = 0.2 - 0.4$ for particles coarser than silt and concentrations > 1% in experiments with very high basal melting rates. Altogether, friction by ice with basal debris is significant (Cohen et al., 2005).

A temperate layer of ice has no significant thermal gradient and cannot conduct heat (see Section 3). Any heat produced at the base therefore causes basal melting, with the resultant melting rate $M_r$ given by:

(2) $\qquad M_r = \dfrac{(Q_{geo} + Q_{fr})}{H_{ice}\, \rho_{ice}}$ in [m s$^{-1}$]

where $H_{ice}$ is the heat of fusion of ice and $\rho_{ice}$ the density of ice. Water pressure $P_w$ and friction coefficient $\mu$ (which is largely controlled by debris concentration) are likely to vary significantly in space and time. These variations affect the frictional heat production (equation 1), and hence the melting rate. The potential contribution of frictional heat due to basal sliding as a function of $\mu$ and $P_w$ is graphically illustrated in Fig. 1. It is clear that under a range of circumstances, frictional heating can be equal or greater than the geothermal heat flow (see also Paterson, 1994; Calov and Hutter, 1994). Even at $\mu = 0.05$ (close to the friction coefficient of Teflon) heat production can be significant. At moderate sliding velocities (<10 m yr$^{-1}$), heat production at a warm base can be twice the heat production at a cold base, whilst at high sliding velocities typical of ice streams (>50 m yr$^{-1}$), frictional heating swamps geothermal heat flow. Note that the likely significant spatial variations in friction coefficient and bed roughness at the base of an ice sheet imply that it is not reliable to derive geothermal heat flow on the basis of basal melt rates alone (cf. Fahnestock et al., 2001; Greve, 2005).

The question now becomes: what happens with the water produced by frictional heating? There are two possibilities, not mutually exclusive:

1) The water drains away, initially along a film which may evolve into a dispersed drainage system and further into a channelized drainage system (e.g., Weertman and Birchfield, 1983). Generally water will drain away in the direction of ice flow and ultimately discharge from the ice sheet, thus representing overall mass loss. Any thermal gradient along the base will be continuously smoothed by the advective heat transport of the flowing water. As a result, the entire basal environment (basal ice, water and top of bedrock) approaches the pressure melting point, with a near-uniform temperature and no significant horizontal thermal gradients.

2) If water cannot drain away freely, the water will remain under pressure. Water may move upwards through the temperate ice layer to the cold-temperate-boundary (CTB). Here it will refreeze and release its latent heat; this heat will warm up the cold ice just above the CTB, and thus thicken the temperate layer, as further explained below.

## 3 Growing and maintaining a temperate ice layer

The growth and continuance of a temperate ice layer below cold ice is an interesting problem in its own right. The problems associated with this are two-fold (Fig. 2):

i) because cold ice above the CTB moves (by internal deformation) towards the margin, there is a strong component of horizontal thermal advection. The isotherms are compressed and the effective thermal gradient at the base of the cold ice just above the CTB is steep, much steeper than can be maintained by heat conduction alone (Dahl-Jensen, 1989; Paterson, 1994; Funk et al., 1994). In Borehole D north of Jakobshavn Isbrae, this thermal gradient has been measured as 0.05 °C m$^{-1}$ (= 50 °C km$^{-1}$!) (Lüthi et al., 2002). There is thus the tendency to cool and hence shrink the temperate layer. For a temperate layer to exist and grow, energy must thus be added to the CTB (e.g., Clarke et al., 1977; Blatter and Hutter, 1990);

ii) a temperate ice layer has no bulk thermal gradient (it has in fact a very small negative gradient, but this is ignored here), so no heat can be conducted through it; it forms a near- ideal thermal barrier (e.g. Aschwanden and Blatter, 2005).

Several mechanisms can be invoked to add energy to the CTB, despite the absence of a thermal gradient (Fig. 2):

1) Bedrock highs can conduct heat: if these penetrate the CTB, heat can be conducted into cold ice. This mechanism is limited by the heat conductivity of rock and the height of the obstacle; it cannot explain a CTB above a bedrock high;

2) Temperate ice layer may locally thicken by internal deformation, i.e. by folding or thrusting where basal ice flow is heterogeneous near obstacles (e.g. Bell et al., 2014). However, this can only redistribute temperate ice and move it into cold ice, rather than lead to an overall thickening of the temperate layer itself.

3) Strain heating within deforming cold ice above the CTB. Given that this zone is subject to high strain this maybe significant (e.g., Clarke et al., 1977; Iken et al., 1993; Cuffey and Paterson, 2010). However, the sharp nick in the temperature profile in Borehole D (Fig. 2, see Lüthi et al., 2002) suggests that most heat is transferred across the CTB, rather than generated by strain heating within the cold ice above the CTB;

4) Transport of water through the temperate layer. If water moves upward through the temperate layer and crosses the CTB, it freezes, releases its latent heat and warms the cold ice just above the CTB. As the ice temperature rises to $T_m$, the CTB moves upwards and the temperate layer thickens. A water flux through the temperate layer therefore equates with a heat flow. An intergranular vein network of water likely exists along grain boundaries in temperate ice (Nye and Frank, 1973; Mader,1992). If (locally) $P_w > P_i$ because of a poorly connected subglacial drainage system (high $P_w$ has been observed in drill hole C in Greenland; Iken et al., 1993) then water can migrate upwards against gravity, either by percolation or possibly by hydraulic fracturing. Water may also migrate as bubbles through ductile deformation, as suggested by the experiments of Wilson et al. (1996). Lovell et al. (2015) suggested that the 'dispersed' basal ice facies found close to the base in surge glaciers may form by shear deformation and partial melting along grain boundaries, resulting in an upward flux of liquid and gas along grain boundaries. Although temperate ice has a low permeability (Lliboutry, 1971), even a small water flux is very effective in transporting 'heat' because of the large latent heat of melting compared to the specific heat capacity of ice: 1 kg of freezing water can heat 160 kg of ice by 1 °C (Paterson, 1994). To maintain the CTB with a thermal gradient of 0.05 °C m$^{-1}$ just above it (Fig. 2), an energy flow of 0.105 W m$^{-2}$ is required, about twice the normal geothermal heat flow (see Appendix). This in turn requires a flux of water through the temperate layer of ~23 mm yr$^{-1}$ (see Appendix), well within the range of water production by frictional melting at moderate to high sliding velocities. This mechanism is probably the most important to grow and maintain a temperate layer.

## 4 The creep component in temperate ice

In Weertman's (1957) model, stress concentrations build up around an obstacle according to:

(3) $\qquad \sigma_{stoss} = \frac{1}{6} \tau \, \lambda^2 (wh)^{-1}$

where $\sigma_{stoss}$ is the normal stress on the stoss side, acting horizontally; $\tau$ is the overall shear stress; $\lambda$ is the spacing between obstacles; and $w$ and $h$ the width and height of the obstacle, respectively. Weertman (1957) assumed that the creep component of ice flowing around a hard obstacle worked with a rheology according to a standard power law, enhanced by stress concentrations around the obstacle. This law ('Glen's flow law') concerns the general relation between imposed deviatoric stress σ and resulting strain rate $\dot{\varepsilon}$:

(4) $\dot{\varepsilon} = A\sigma^n$

where $A$ is a flow parameter and $n$ the creep exponent. Stress and strain rate are tensors; in equations (4) and (5) their second invariants, or the effective stress and strain, are implied (e.g. Cuffey and Patterson 2010). If $n > 1$, the flow is non-Newtonian. The flow parameter $A$ is strongly dependent on temperature; this temperature dependence follows the Arrhenius relation, so that the relationship between strain rate, deviatoric stress and temperature is typically described as:

(5) $\dot{\varepsilon} = A_0 \sigma^n e^{(-\frac{Q_a}{RT})}$

where $A_0$ is a constant, $R$ the gas constant, and $Q_a$ the activation energy (Glen, 1955; Alley, 1992; Cuffey and Paterson, 2010). For ice, experiments broadly suggest that $n \sim 3$ and $Q_a \sim 60$ kJ mol$^{-1}$ for $T < $ -10 °C and $Q_a \sim 120$ kJ mol$^{-1}$ for $T >$-10 °C (e.g., Barnes et al., 1971; Weertman, 1983; Duval et al., 1983; Alley, 1992; Cuffey and Paterson, 2010). With these values it is termed here the 'standard power law'; such a standard power law is, with different parameter values, widely

applicable at appropriate conditions to many crystalline materials such as quartz, olivine and metals (e.g., Poirier, 1985). Comparisons with borehole tilt deformation studies suggest that the standard power law describes the rheology of clean cold ice reasonably well (e.g., Dahl-Jensen and Gundestrup, 1987; Lüthi et al., 2002). Note that strain rate $\dot{\varepsilon}$ increases exponentially with temperature. The rate-controlling deformation mechanism of standard power-law creep in ice and other crystalline materials such as quartz and olivine is normally regarded to be intracrystalline creep, mainly by dislocation glide

along basal planes (e.g., Duval et al., 1983; Poirier, 1985; Alley, 1992). The question is whether temperate ice behaves according to the same standard power law, has the same rate-controlling deformation mechanism (e.g., Hooke, 1981; Parizek and Alley, 2004) and is applicable to temperate ice flow around hard obstacles.

Firstly, experimental data compiled by Morgan (1991), all performed under constant stress (100 kPa), are plotted in Fig. 3 to

illustrate the effect of temperature on the strain rate. The natural logarithmic of strain is plotted against the reciprocal of temperature, so that a straight line would confirm the Arrhenius relations within the standard power law (Equation 5). For temperatures between -5 and -0.5 °C, the data plot on a straight line, the gradient of which equals ($-Q_a/R$), confirming the Arrhenius relation in equation (5). However, at about -0.02 °C there is a sharp nick in the trend, with strain rates increasing by up to a factor 5 to 10 as the melting temperature is approached. Thus, at constant stress, ice above c. -0.2 °C shows a

sudden weakening, evidently not described by the Arrhenius relation of equation (5), suggesting that a standard power law cannot be used to model temperate ice and potentially suggesting a switch in dominant deformation or accommodating mechanism (see also Barnes et al., 1971; Colbeck and Evans, 1973; Morgan, 1991).

Secondly, the value for the creep exponent $n$ in for temperate ice is highly uncertain. Experiments at constant temperature

but varying stress (T $\sim$ -0.01 °C , $\sigma$ = 10-100kPa) by Colbeck and Evans (1973) suggest $n = 1.3$; experiments of creep at the melting temperature past a sphere by Byers et al. (2012) also suggest $n < 1.5$. By analysing bulk stress and strain rate using borehole tilt measurements in a 3D borehole network in the temperate Worthington Glacier, Marshall et al. (2002) noted $n \sim$ 1 at low stresses, but a change to $n \sim 4$ at higher stresses (>1.8 kPa). De La Chapelle et al. (1999) noted a similar change

from $n \sim 1.8$ at low stresses  to $n \sim 3$ at high stress (>250 kPa)  in experiments of pure ice in the presence of a brine (i.e. not temperate ice *sensu stricto*).   Analysis of bulk stress and strain rate of the temperate Glacier de Tsanfleuron, however, suggest $n \sim 1$ (Chandler et al., 2008).  Note that if $n \sim 1$, the stress-strain relation becomes  a linear law rather than a power law, and represents Newtonian viscous flow.   There is further uncertainty as there appears to be a discrepancy between temperate ice behaviour seen in  laboratory experiments and in temperate glaciers (Cuffey and Paterson, 2010).

It thus appears that deforming temperate ice behaves fundamentally different from deforming cold ice and cannot be reliably modelled with a standard power law.  The sharp transition just below the melting temperature suggests that this difference is largely related to the presence of water.  Duval (1977) noted  in temperate ice experiments that a rise in water content up to c. 0.8% leads to a 5-8 times strain rate increase.  Temperate ice in Alpine glaciers can contain 1-2 % water (Vallon et al., 1976), which has been observed in experiments to gather along grain triple junctions (Barnes and Tabor, 1966; Wilson et al., 1996), so that it is likely that a vein network along triple junctions exists (Nye and Frank, 1973; Mader, 1992).  Partial melting of a deforming  temperate layer is furthermore suggested by the formation of bubble-free ice, both in experiments (Barnes and Tabor, 1966) as well as in Alpine and surging Svalbard glaciers (Tison and Hubbard,  2000, Lovell et al., 2015). The dominant deformation mechanism for temperate ice, however, is uncertain and it is possible that different deformation mechanisms operate simultaneously.  Possible deformation mechanisms and their potential enhancement by the presence of water are suggested below.

1)   Diffusion creep is enhanced by the presence of liquid along grain boundaries, since that liquid functions like a fast diffusion path (Pharr and Ashby, 1983; Raj, 1982; Goldsby and Kohlstedt, 2001).

2)   Dislocation creep is also enhanced by liquid (Duval, 1977; De la Chapelle et al., 1995; 1999).  Water along grain boundaries decreases the surface area of grain-to-grain contacts and cause an increase in grain-to-grain contact stresses; this will enhance dislocation creep (De La Chapelle et al., 1999) but also other deformation  mechanisms. Liquid may also suppress strain hardening and enhance easy intracrystalline basal slip (Duval, 1977;  De La Chapelle et al., 1999).

3)   Dynamic recrystallization and grain growth is rapid in deforming temperate ice (e.g., Duval et al., 1983; Wilson, 1986).  Dynamic recrystallization aids dislocation creep as it grows crystals with orientations favourable for easy basal slip and suppresses strain hardening (e.g. Duval et al., 1983).  Dynamic recrystallization results in a coarse grain size and should aid development of a crystal fabric.

4)   Grain boundary sliding is commonly invoked to explain weakening in ductilely deforming materials (superplasticity).  Superplasticity has been experimentally achieved in ice at very low temperatures (-30 to -80 C°) and very small (3-40µm) grain sizes (Goldsby and Kohlstedt, 1997, 2001; Goldsby and Swainson, 2005), markedly different to the temperate ice under discussion here.  Whether grain boundary sliding in ice leads to the formation or

destruction of a crystallographic fabric appears debatable (Goldsby and Kohlstedt, 2001; 2002; Duval and Montagnat, 2003; Goldsby and Swainson, 2005).

5) Grain boundary melting (or 'internal pressure melting') has been observed in ice deformation experiments with indentors by Barnes and Tabor (1966), Barnes et al. (1971) and Wilson et al. (1996). The principle is that ice melts at highly stressed grain boundaries (Fig. 4) and liquid is transported to lesser stressed grain boundaries where it refreezes – or it may escape if the intergranular vein network is efficient. Either way, this leads to strain. For ice (in contrast to almost all other materials), it is important to emphasise that grain boundary melting involves a negative volume change upon melting ($\Delta V = -9\%$), which makes grain boundary melting under elevated stresses a thermodynamically favourable mechanism. The distance of heat transport to the stressed grain boundaries necessary to sustain grain boundary melting is half the grain size (Fig. 4), some three orders of magnitude smaller than the size of most bedrock obstacles. This mechanism only operates if ice and water coexist in thermal and chemical equilibrium, and would thus not be observed in experiments with cold ice and where the liquid is a brine (cf. De La Chapelle, 1995; 1999). Grain boundary melting is supported by the formation of bubble-poor ice at the base of temperate glaciers: both Tison and Hubbard (2000) and Lovell et al. (2015) show that such ice is not formed by direct freeze-on (regelation ice), but by a metamorphic process involving partial melting. Grain boundary melting is loosely analogous to pressure solution (solution-precipitation creep) observed in salts and limestone, insofar that material changes from solid to liquid or vice-versa along grain boundaries in different stress states (Pharr and Asby, 1983; McClay, 1977; Rutter, 1983), but differs in that grain boundary melting creates its own liquid.

Which of these deformation mechanisms is dominant is difficult to establish. Tison and Hubbard (2000) documented large grain sizes (5-20 mm) and a well-developed crystallographic fabric in deforming ice at the base of the temperate Glacier de Tsanfleuron, features not compatible with grain boundary sliding as a dominant deformation mechanism. Diffusion creep, grain boundary sliding and grain boundary melting all work on grain boundaries and are grain-size sensitive: they are favoured by a small grain size and the presence of a liquid; these mechanisms normally result in $n < 2$, and thus could explain the $n \sim 1$ behaviour seen in some experiments and natural glaciers. However, all grain-size sensitive mechanisms are at odds with the large grain sizes observed and can, on their own, not explain well-developed fabrics. Well-developed fabrics potentially attest to dislocation creep, but this is in turn at odds with the $n \sim 1$ behaviour commonly observed. Altogether there is no clear evidence of a single dominant deformation mechanism, and all deformation mechanisms mentioned above may contribute. Considering the near-unique pressure-melting behaviour of $H_2O$, grain boundary melting is worthy of further study. The change in the stress-dependency as observed in the Worthington Glacier (Marshall et al., 2002) as well as in some experiments (De La Chapelle et al., 1999) suggests that the dominant deformation mechanism in temperate ice depends on the magnitude of stress. For the moment the rather non-generic term '*melt-assisted creep*' is used herein. A strong crystallographic fabric and concentrations of dust or silt particles are known to significantly weaken cold

ice in simple shear (e.g., Lile, 1978; Paterson, 1984, 1991; Dahl-Jensen and Gundestrup, 1987; Azuma, 1994), but whether this leads to further weakening of temperate ice is not known. There is still much unknown about creep in temperate ice; regardless of the actual mechanism or the precise flow law, all experiments suggest that temperate ice is significantly weaker than, and behaves very differently from, cold ice.

## 5 The pressure melting component

In the classic Weertman's model, the stress concentration on the stoss side (Fig. 5a) that build up according to equation (3) causes a lowering of the melting temperature $\Delta T_m$ according to:

(6) $\qquad \Delta T_m = -C\sigma_{stoss}$

where C is the pressure melting constant ($7.4 \cdot 10^{-8}$ K Pa$^{-1}$). As an example, with $w = 1$ m, $h = 1$ m and $\tau = 124$ kPa, similar to the parameters used by Weertman (1957), this would result in a stoss-side normal stress $\sigma_{stoss} = 330$ kPa (see also Appendix), causing a lowering of the melting point of $\Delta T_m = -0.025$ °C. Assuming the deviatoric stress on the lee-side is equal but opposite, the melting temperature at the lee side is higher by an equal amount, so that $\Delta T_m = +0.025$ °C. Weertman (1957) envisaged that water freezing on the lee-side released latent heat; this excess heat was regarded to be *transported through the obstacle* towards the stoss side where it caused further melting. The resultant heat flux through the obstacle $Q_{ob}$ is then controlled by the total temperature difference between the lee and stoss side ($2\Delta T_m = 0.05$°C in this example), the thermal conductivity of rock $K_r$ and the length $l$ of the obstacle :

(7) $\qquad Q_{ob} = K_r 2\Delta T_m l^{-1}$

This heat flux was regarded to control the amount of thermal energy available at the stoss side for melting to proceed and hence as controlling the velocity by melting $V_{pm}$ :

(8) $\qquad V_{pm} = Q_{ob} H_{ice}^{-1} \rho_{ice}^{-1} = K_r 2\Delta T_m l^{-1}H_{ice}^{-1} \rho_{ice}^{-1} \propto l^{-1}$

where $\rho_{ice}$ the density of ice. This model has a number of problems. Firstly, if cavitation occurs it is unclear how heat can be transported to the stoss-side (as pointed out by Lliboutry, 1993). Secondly, $V_{pm}$ is inversely proportional to the length of the obstacle, following equation (8). This implies that ice melting around an obstacle that is, say, four times longer than another obstacle (Fig. 5b), would be four times slower, even though this obstacle is more streamlined (having a longer aspect ratio). This result contradicts most observed geomorphology (Stokes and Clark, 1999; Bradwell et al., 2008) and supports the notion that pressure melting is not dominant for large obstacles.

## 6 Stoss-side pressure melting in temperate ice

What are the rate controlling factors for stoss-side pressure melting in a layer of temperate ice? In the conceptual model here (Fig. 5c), the temperate layer is thicker than the height of the obstacle. Water is continually produced by frictional heating,

there is a net-melting environment with an excess of water, and water pressure on the ice-bed contact will be high. Depending on the basal melt rate and the amount of water flowing along the ice-rock interface, heat advection by flowing water may well be more efficient than heat conduction through rock or ice. In that case no significant thermal gradients through bedrock obstacles can build up and the entire basal system (temperate ice, water, and top rock) is kept at thermal

equilibrium at $T_m$. The only exception is the stoss-side of a bedrock obstacle, where the melting temperature is continually depressed as a result of the concentrated deviatoric stress acting onto it (Fig. 5c). Thus, the problem of stoss-side pressure melting is reduced to a single 'cool spot' at the stoss side, with a temperature $\Delta T_{stoss}$ below the ambient $T_m$ ($\Delta T_{stoss}$ = -0.025 °C in the example, with similar parameter values as Weertman, 1957). To sustain stoss-side pressure melting, heat needs to be transported only a short distance towards this cool spot: anywhere else is at $T_m$. Although total $\Delta T$ is half compared to the

Weertman model (there being no 'warm spot' at the lee side), the transport distance is much smaller. Most obstacles will be wider than high, in which case the critical transport distance is 0.5 $h$. As a consequence the heat flow towards the stoss side will be greater than in Weertman's model, and it is independent of the length of the obstacle. Cavitation may occur, as this system is not dependent on regelation. The process can work without any regelation in an overall melting environment, which is compatible with the observation of continuous net basal melting at the base of ice sheets (e.g., Fahnestock et al.,

2001). If regelation occurs on the lee side (because of lower stress) of one obstacle, any excess heat will be advected by flowing water to the stoss-side of the *ne*xt obstacle. In this process, the rate controlling factors are the height of the obstacle (but not the width or the length) and the efficiency of the heat advection by flowing water. The process is *not* limited by heat flow through the obstacle (cf. Weertman, 1957; Kamb, 1970), and the length of the obstacle becomes irrelevant. This type of stoss-side melting will be faster than in Weertman's model for all but the shortest obstacles, and certainly so for obstacles

with $l>h$, which is the case for most observed bedrock bedforms in even the roughest of deglaciated terrains (Bradwell, 2013; Roberts and Long, 2005).

## 7 Effect of surface water input on temperate ice on a rough bed

Influx of surface melt water represents addition of thermal energy to the basal environment and can further aid stoss-side melting and hence basal motion. Consider the dramatic influx of surface melt water by the sudden drainage of supraglacial

meltwater lakes in West Greenland (e.g., Das et al., 2008). This water is relatively warm (c. 1° C; Tedesco et al., 2012) and such an influx thus represents a significant addition of thermal energy to the base. Sudden influx of warm melt water may have the following consequences:

a) Increase of basal water pressure $P_w$, resulting in a drop in effective pressure $P_e$, lowering the friction on flat surfaces. Frictional heating and drag on the flats will drop, as long as $P_w$ remains high. On the other hand, because

there is less drag on the flat surfaces, the normal stress $\sigma_{stoss}$ onto the stoss side of obstacles, increases (also temporarily), enhancing stoss-side melting as well as creep.

b) The basal system is flushed with water that is well above the ambient $T_m$. Given the very fast recorded flow of large amounts of water (Das et al., 2008), it is assumed here that little heat is lost during englacial transport. Water entering the basal system at 1 °C is 1.5 °C above $T_m$ and 1.525 °C above the stoss-side melting point. There is thus potentially a steeper thermal gradient between the warm water and the cold spot at stoss-side and more thermal energy is available.

This may lead to accelerated stoss-side melting, which would continue until all the water temperature has cooled to $T_m$.

c) A more longer-term effect is that if warm water is added to the base and cannot drain away freely, the additional thermal energy will lead to a thickening of the temperate layer (upwards migration of the CTB), in essence cryo-hydrological warming of the basal system (cf. Phillips et al., 2010).

The rate controlling factors of this enhanced stoss-side melting are (i) the flux of surface melt water; (ii) the temperature of

this water, (iii) the dissipation of the extra heat, for instance by further melting of ice above the flat surfaces.

In West Greenland, sudden supraglacial meltwater drainage events are accompanied by (a) an immediate (hour time scale) speeding up of surface velocity, with a total horizontal displacement of < 1 m and vertical uplift of the ice surface on a centimetre scale, followed by (b) a longer period (days) of decelerating but still above-average ice velocity (e.g., Das et al., 2008; Shepherd et al., 2009; Hoffman et al., 2011). The centimetre-scale ice uplift is clearly insufficient to lift basal ice over

1-10 m high obstacles that are likely to exist at the base of the Greenland ice sheet, given its gneiss-dominated bedrock (Roberts and Long, 2005; Krabbendam and Bradwell, 2014). The sudden, short jump is probably due to true sliding as basal ice is pushed higher onto (but not over) sloping obstacles due to an increase of $P_w$. However, the longer-term (days) increase in ice velocity may well be caused by accelerated stoss-side melting as described above. It is remarkable that the dramatic lake drainage events have a rather muted effect on ice velocity, strongly suggesting that the basal ice in West Greenland is

'stuck' on the stoss sides of pronounced bedrock obstacles. The term 'lubrication' (e.g., Parizek and Alley, 2004; Shannon et al., 2013) is inappropriate to describe the latter process: it is enhanced pressure melting-regelation, rather than a lowering of friction, that leads to the speed-up.

## 8 Critical obstacle size

Weertman (1957) introduced the notion of the 'critical obstacle size'. Because pressure melting and enhanced creep have

different dependencies on the obstacle size, melting and regelation should be the dominant mechanism for small obstacles, whereas enhanced creep should be dominant for larger obstacles. For his chosen parameters, Weertman arrived at a critical obstacle size of ~ 1 cm (Fig. 6). Replacing the controlling mechanisms with those for temperate ice, but otherwise applying the same parameters, this would result in approximately a 5-10 times increase in creep velocity and a doubling of the pressure melting velocity, because the controlling height is 0.5h. This means that the critical obstacle size becomes even

smaller, e.g. c. 0.5 cm (Fig. 6). However, in subglacial observations and experiments Kamb and La Chapelle (1964) noted that melting/regelation was dominant at much larger obstacles and suggested a critical obstacle size of about 1 metre. They

suggested that whilst the qualitative idea of a critical obstacle size of Weertman (1957) was correct, the quantification was incorrect. In subglacial experiments under high vertical melt-rate conditions, however, Cohen et al. (2000) also suggested critical obstacle size of c. 5 cm. On the scale of typical bedrock obstacles such as roche moutonnées ($l \sim$ 5-50 m, $h \sim$ 1-10 m), enhanced creep will be dominant and, as discussed above, will in temperate ice be controlled by some form of melt-assisted creep. Pressure melting may not be important for bedrock obstacles on a rough bed, but can be important for ice flow around cobble-sized debris.

## 9 Discussion

### 9.1 Summary of rate-controlling mechanisms

In summary, ice flow around a bedrock obstacle in temperate ice is accommodated by stoss-side pressure melting or by enhanced creep, with creep being more important for larger obstacles. In temperate ice, the enhanced creep component of basal motion operates close to or at the melting temperature. The rheology of this melt-assisted creep is poorly constrained, but does not behave according to standard power-law creep and is up to one magnitude faster than creep in cold ice, and may involve a component of grain boundary melting. The effects of strain softening and dust particles are uncertain; more laboratory experiments on the deformation of temperate ice may help to better understand creep in temperate ice. If a temperate layer exists that is thicker than the height of bedrock obstacles, it is proposed here that stoss-side pressure melting is constrained by:

- Stress concentration on the stoss-side, which depends on the surface area of the stoss-side with respect to the spacing of obstacles, and also to the slope of the obstacles, something not considered here;
- Height of the obstacle, which is critical to the transport distance of heat;
- Frictional heating on the flat surfaces, which causes the production of excess meltwater;
- The efficiency and possible localisation of the local drainage network;
- Input of surface meltwater.

In contrast to the classic Weertman model, stoss-side pressure melting is *not* constrained by heat flow through the rock obstacle, cavitation is possible, and the length of the obstacle is not relevant.

### 9.2 Basal sliding regimes throughout an ice sheet

The proposed model has implications for ice sheet behaviour and the modelling thereof. Consider a hypothetical half-ice sheet (Fig. 7), based on the thermo-mechanical model by Dahl-Jensen (1989) and with a passing resemblance to the Greenland Ice Sheet. Thermal gradients are strongly affected by horizontal motions as ice velocity exceeds the rate of heat

conduction, so that a cold 'tongue' occurs within the ice sheet (e.g., Dahl-Jensen, 1989; Iken et al., 1993). The bed is regarded as rough and hard. In such a model, three thermomechanical basal regimes can occur, with a potential fourth operating seasonally.

1) In the cold-based regime, the thermal gradient crosses $T_m$ well into bedrock. All geothermal heat is conducted upwards through the ice. No sliding occurs, and frictional heating is zero. All deformation is internal and can be described by power-law creep.

2) At some point the thermal gradient crosses $T_m$ at the base of the ice sheet, and the base is at the pressure melting point. Sliding starts and frictional heating kicks in. In this regime, the friction coefficient will be highly variable as some patches will be frozen, with very high friction coefficient (Barnes et al., 1971; Budd et al., 1979; Zoet et al., 2013) but low sliding velocities, whereas other patches will be wet, with lower friction coefficient but higher sliding velocities. Excess heat may still be conducted away by the overlying cold ice and water may well regelate onto lee-sides and on overlying ice, thus limiting the amount of water present. In this regime, basal sliding by some form of classic Weertman sliding is likely. Almost all creep still occurs in cold ice and can be described by a standard power law.

3) Continuous frictional heating overwhelms the heat conducting capacity of the overlying ice and a thick temperate layer develops. The critical thickness is probably reached when the CTB is higher than most hard obstacles, and is thus a function of basal roughness. Where or when this critical thickness is reached depends in part on the thermal structure of the ice sheet, and in particular the thermal gradient near the base: a steeper gradient requires more frictional heating to develop a temperate basal layer. Once a thick temperate ice layer has developed, basal motion occurs according to the processes described above: (i) on the flat surfaces frictional sliding occurs; (ii) ductile flow will occur by melt-assisted creep, much faster than standard power-law creep; (iii) stoss-side melting will be faster than in Weertman Sliding.

4) if large amounts of surface meltwater can drain to the base of the ice sheet, for instance by periodical drainage of supraglacial lakes, a different temporal thermo-mechanical regime develops. Influx of surface meltwater adds thermal energy to the basal environment; this thermal energy is available in part to further accelerate stoss-side pressure melting.

## 9.3 Relevance for ice streaming and ice sheet modelling

The corollary of the processes described herein is that if a thick temperate layer is present, basal motion over a hard bed with bedrock humps provides less drag than ice modelled with cold ice properties. This finding is relevant to ice-sheet modelling. For instance, Peltier et al. (2000) argue that the Laurentide ice Sheet cannot be adequately modelled using a standard power-law rheology; instead they suggest a different, weaker bulk rheological behaviour. This solution, however, is at odds with findings that the bulk rheology of ice in boreholes can be adequately described as standard power-law creep. Instead, it may be more realistic to invoke fast, weak basal motion, even on hard beds, in ice-sheet models. How may this relate to ice-streaming mechanisms? A thick temperate layer has been observed in boreholes adjacent to the Jakobshavn Isbrae and

inferred within its centre (Lüthi et al., 2002), whereas boreholes in non-ice streaming parts of the Greenland ice sheet show an absence of a temperate layer (Ryser et al., 2014). Ice streams are widespread and their locations appear to be controlled by a range of factors, of which topographic steering and the presence of soft, deforming sediment bed are seen as the most important (e.g., Cuffey & Paterson 2010; Winsborrow et al., 2010). There are, however, numerous palaeo-ice streams that neither portray strong topographic channelling, nor have soft-sediment (till) at their base (Bradwell et al., 2008; Bradwell, 2013; Eyles, 2012; Krabbendam et al., 2016).

A modern example of such an ice stream maybe the Northeast Greenland Ice Stream (NEGIS) (Joughin et al., 2001; Fahnestock et al., 2001). While there is topographic steering near its outlet glaciers, topographic steering is weak over much of its length (Joughin et al., 2001; Christianson et al., 2014). Seismic and radio-echo-sounding have shown the base of the NEGIS to be rough (e.g. Christianson et al., 2014) and also suggests the possible presence of water-rich till. However, the presence of soft, deformable till below the NEGIS is not proven and, if existent, may not be the rate-controlling factor, as deforming till will not help in moving ice over bedrock humps. The bedrock geology of Greenland is dominated by Precambrian gneisses, and such rocks almost certainly underlie most of the inland track of the NEGIS. Deglaciated areas of similar Precambrian gneisses in Canada, Scotland, Scandinavia and west Greenland generally show a lack of till, extensively exposed bedrock and a rough landscape of rock knolls and rock basins (Roberts and Long, 2005; Krabbendam and Bradwell, 2014). On these grounds, deformable till beneath the NEGIS is unlikely. Using high-resolution radar profiles to reconstruct an isochron stratigraphy, Fahnestock et al. (2001) reported large areas of 'missing' basal ice below the NEGIS, and attributed this to very high, long-term basal melt rates (up to c. 100 mm yr$^{-1}$). These high basal melt rates were thought to be caused by anomalously high (10 times above normal values) geothermal heat flow. The geophysical evidence for such anomalously high geothermal flow, however, is localised and non-unique, and does not cover the track of the NEGIS. More importantly, the contribution of frictional heating to high basal melting rates was not explicitly taken into account by Fahnestock et al. (2001). Instead, I suggest here that the NEGIS may possess a substantial basal layer of temperate ice, formed and maintained largely by frictional heating. Basal melting rates of 100 mm yr$^{-1}$ are possible in areas of high friction and/or high ice velocity (Fig. 1). In the inversion technique employed by Joughin et al. (2001), a weak basal temperate ice layer would potentially give a similarly low basal drag as a soft, deformable bed.

The occurrence of a temperate basal layer below cold ice is potentially an important factor controlling ice dynamics and ice streaming. However, the extent and thickness of such a layer is at present only constrained at a few widely dispersed boreholes. The occurrence of a basal temperate layer and estimates of water content can be detected remotely by using radio-echo sounding at certain frequencies (Björnsson et al., 1996; Murray et al., 2000, 2007), and it would be interesting to see how thick and how widespread temperate basal layers are in modern-day ice sheets. The uncertainty of the rheological behaviour and the exact form of the relevant flow law for temperate ice means that, unfortunately, ice bodies with significant

amounts of temperate ice cannot be modelled reliably at present, except by purely empirical constraints. Arguably, there is a need for more laboratory and natural experiments to better understand the mechanical behaviour of temperate ice.

Future challenges to improve constraints and parameterisation of dynamic ice sheet modelling include a better knowledge of the rheology of temperate ice, basal friction and frictional heating, basal roughness and distribution and thickness of the basal temperate layer.

## 10 Conclusions

Basal motion of ice past hard-bed obstacles involves a competition between stoss-side melting and enhanced creep. In a basal layer of temperate ice, stoss-side melting is not controlled by heat flow through the obstacle, but instead by the thickness of the temperate layer, the availability and flux of basal meltwater and the height of the obstacle. Creep in temperate ice is up to ten times faster than in cold ice, suggesting a switch in deformation mechanism to melt-assisted creep. Melt-assisted creep probably comprises several deformation mechanisms triggered or enhanced by the presence of water: grain boundary melting, fast dynamic recrystallisation and enhanced dislocation creep. Together, this suggest that basal motion in temperate ice over a rough, hard bed provides low drag, allowing the possibility of fast ice flow over hard, rough beds. Three different thermo-mechanical regimes control basal sliding: (i) cold-based regime, (ii) warm-based but with a thin temperate layer and (iii) warm-based with a substantial temperate layer. The onset zones of (palaeo)ice streams may coincide with a minimum thickness of the temperate layer, and a thick basal temperate ice layer can explain ice streaming over rough, hard beds, possibly including the Northeast Greenland Ice Stream.

**Acknowledgements**

Doug Macayeal, Martyn Drury, Roderick van der Wal, Steve Roberts and Phil Sargeant are thanked for encouraging and interesting discussions. Sam Roberson is thanked for comments on a previous version of the manuscript. Denis Cohen, Maurine Montagnat and an anonymous reviewer provided critical but very helpful reviews. Doug Benn and Olaf Eisen are thanked for their helpful short comments. This paper is published with the permission of the Executive Director, British Geological Survey.

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

**Figures**

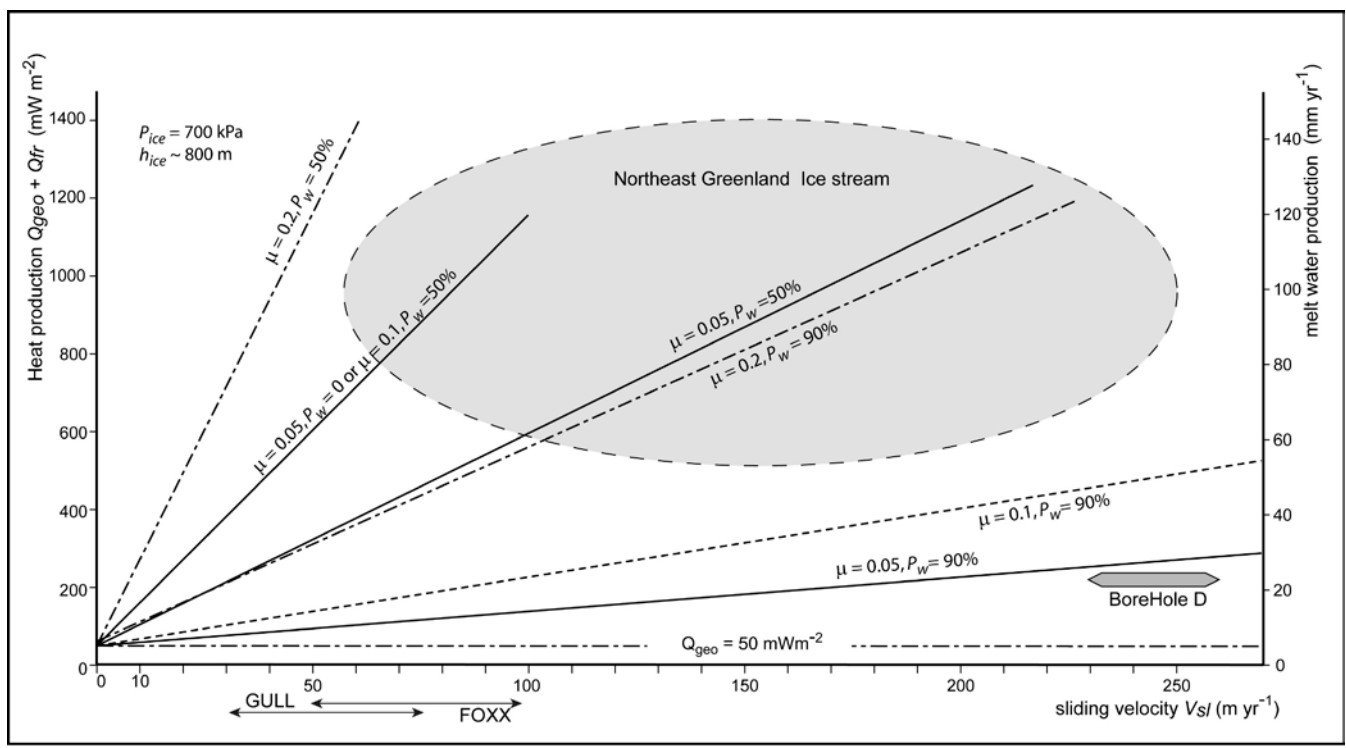

**Figure 1**. Basal heat production (left) caused by geothermal heat flux and frictional heating as a function of basal sliding velocity, for different values of friction coefficient $\mu$ and water pressure $P_w$ as a percentage of overburden pressure. Geothermal heat flow taken at 50 mWm$^{-2}$. On the right hand side the rate of melt water production, assuming all heat is taken up by melting. Sliding velocities of Borehole D (adjacent to Jakobshavn Isbrae) after Lüthi et al. (2002); melt rate calculated on basis of thermal gradient, using equation (1). Sliding velocities and basal melt production of Northeast Greenland Ice stream after Joughin et al. (2001) and Fahnestock et al. (2001). GULL and FOXX show seasonal range of basal ice velocities deduced from borehole data near Swiss Camp, West Greenland (Ryser et al. 2014).

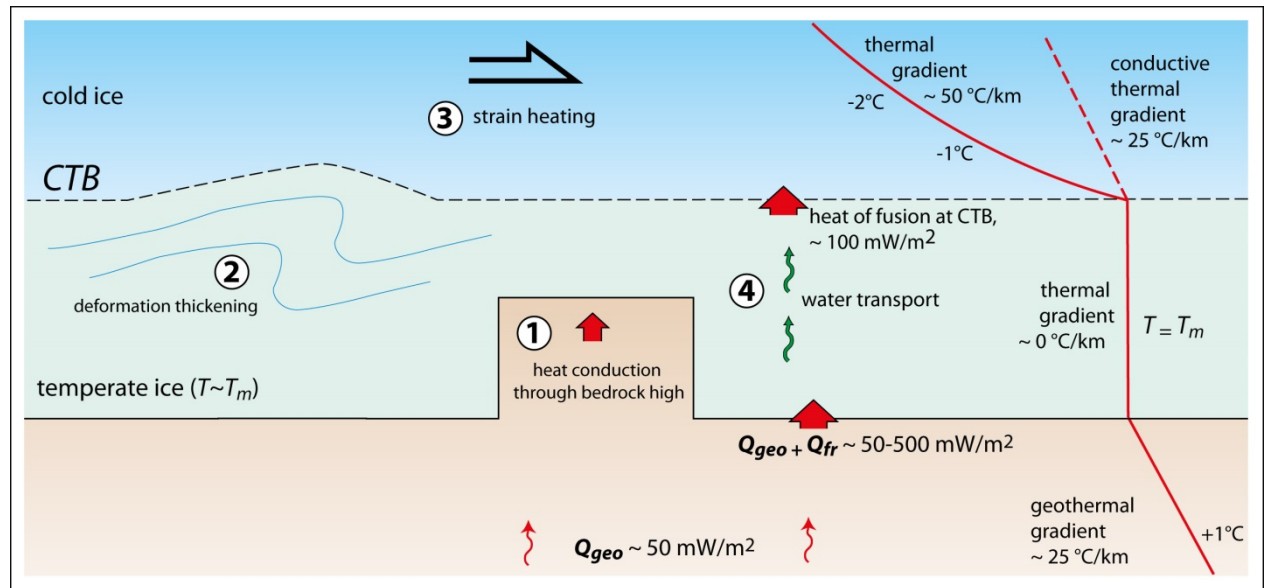

**Figure 2**. Constraints on the thermal growth of the temperate layer, and mechanism for heating cold ice above the CTB; numbers refer to text. Schematic thermal gradient of Borehole D is indicated in red line (after Lüthi et al., 2002). CTB = cold-temperate boundary; $Q_{geo}$ = geothermal heat flow; $Q_{fr}$ = heat production by frictional heating.

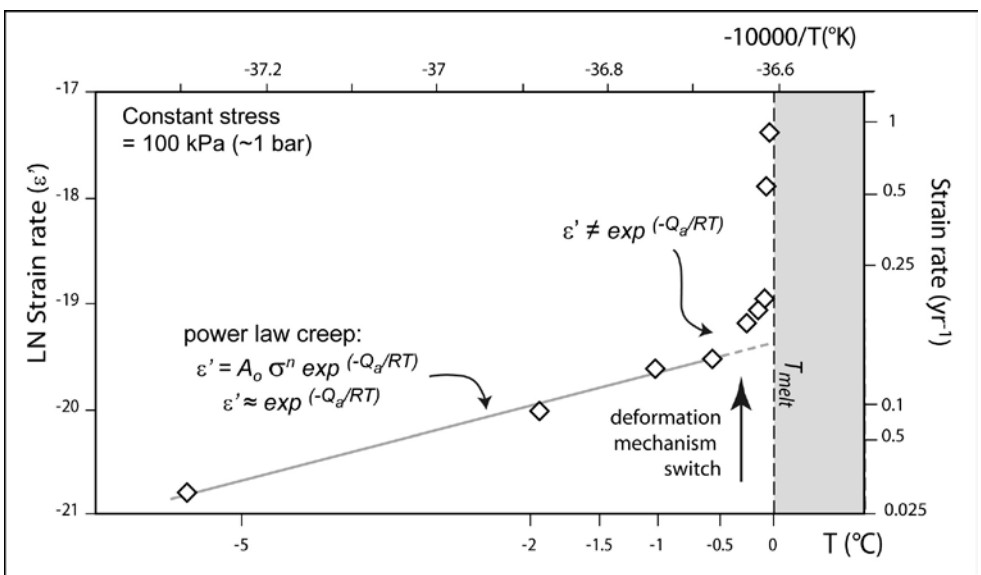

**Figure 3**. Strain rate against temperature, for experiments performed at 100 kPa, replotted after Morgan (1991). X-axis: reciprocal of temperature; Y-axis: natural logarithm of strain rate. Points following the Arrhenius relation within the power

10    law should appear on a straight line.

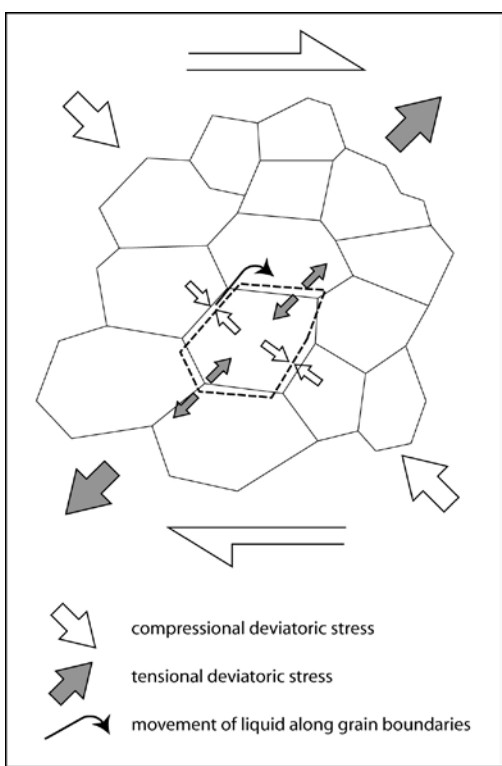

**Figure 4.** Schematic illustration of grain boundary melting under simple shear.  Melting occurs at grain contacts under high stress (compressional deviatoric stress); regelation may occur  at grain contacts under low stress (tensional deviatoric stress). Liquid water moves along grain boundaries.

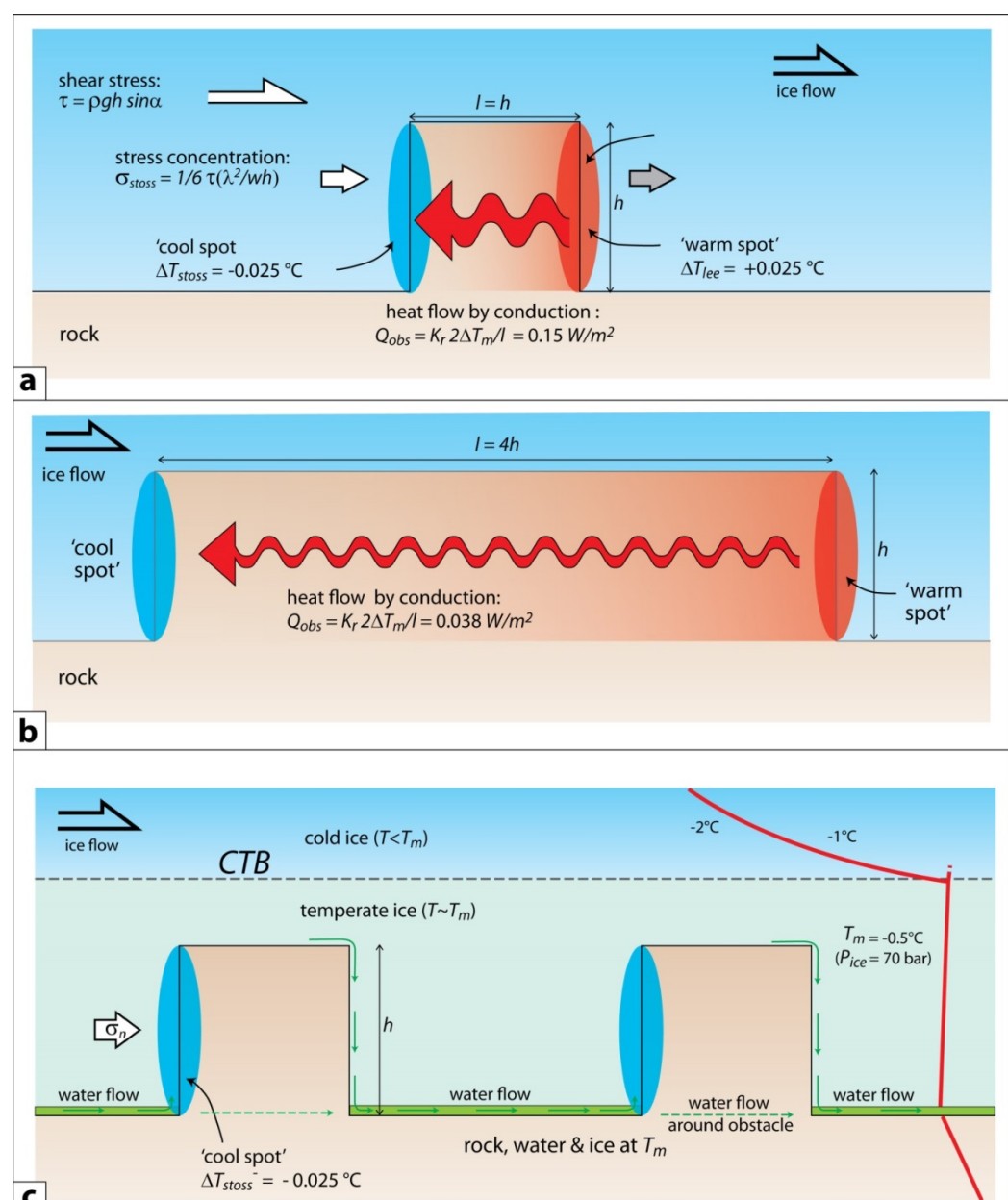

**Figure 5**. (a) Basic Weertman sliding model, illustrating components of pressure melting. Thermal gradient through bedrock obstacle indicated by red arrow; (b) Weertman sliding pressure melting with an elongate obstacle, all other parameters are the same; (c) Pressure melting, water and heat transport in a temperate basal layer with significant meltwater flow: a thermal equilibrium occurs everywhere by heat advection by flowing water except at the 'cool spot' of the stoss side. CTB = cold-temperate ice boundary. Schematic thermal gradient of Borehole D is indicated (after Lüthi et al., 2002). See Appendix for calculations of values in the figure.

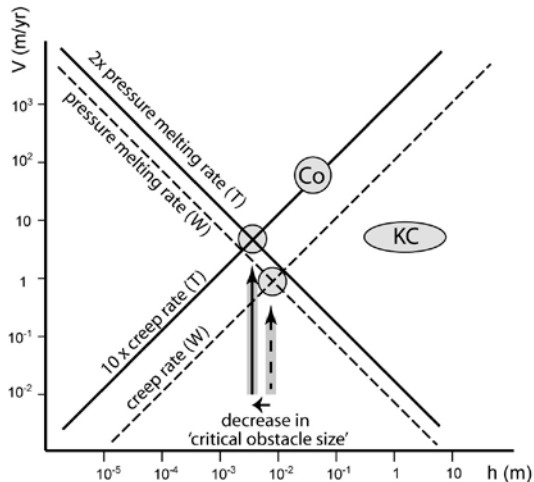

5    **Figure 6.** Sliding velocity due to pressure melting and enhanced creep (logarithmic scale) as a function of height of obstacle (Weertman, 1957). Intersection represents the critical obstacle size. (W) = velocities following Weertman; (T) = velocities in temperate ice; KC, Co = velocity and critical obstacle size of estimates of Kamb and LaChapelle (1964) and Cohen et al. (2000) respectively.

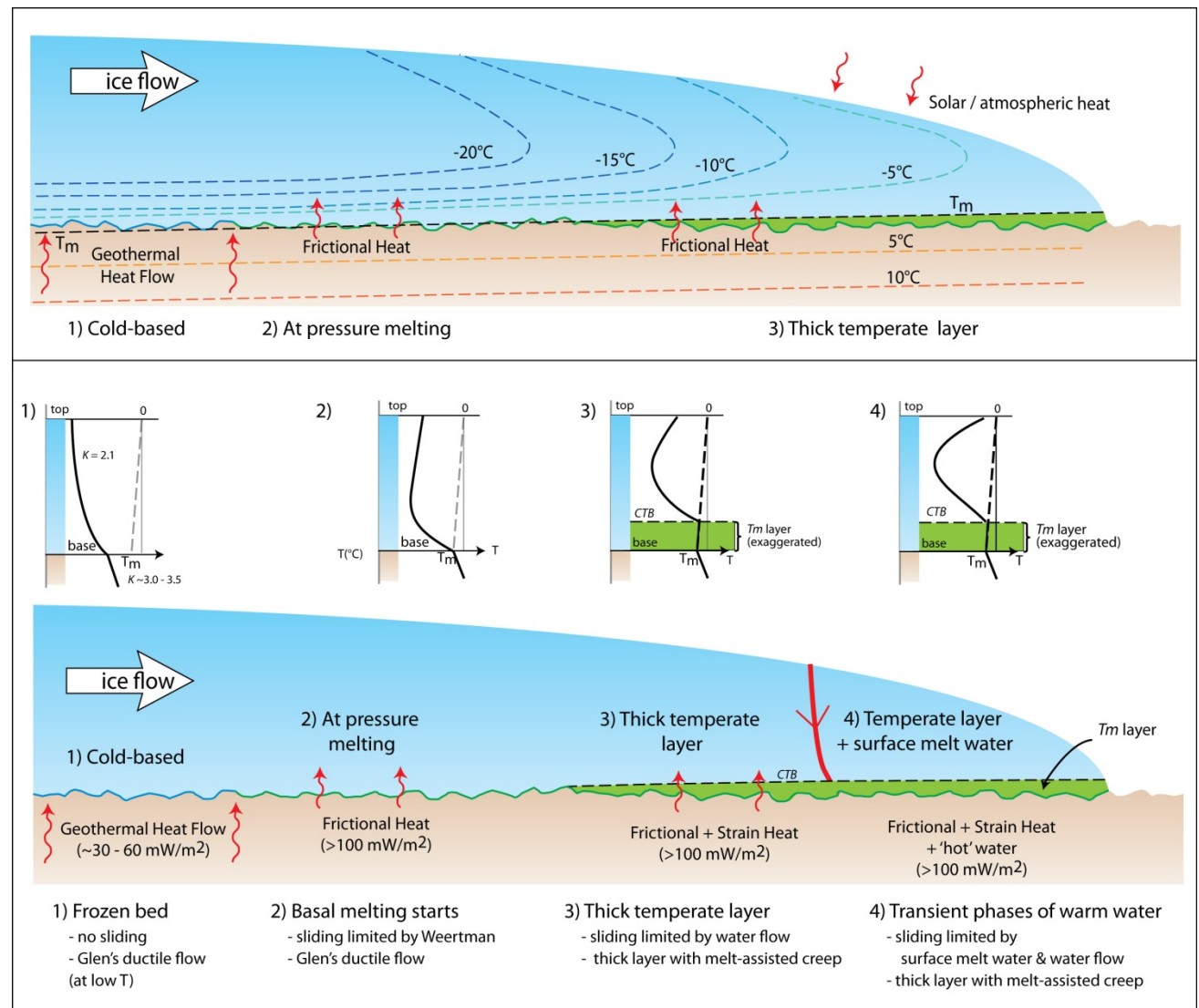

**Figure 7.** Hypothetical half-ice-sheet (e.g., Dahl-Jensen 1989), with different thermal regimes, further explained in the text. *CTB* = cold-temperate boundary; $K$ = thermal conductivity, in Wm$^{-1}$K$^{-1}$, $T_m$ = melting temperature. Numbers correspond to different thermal regimes described in text.

**Table 1**

| Constant | Symbol | Value |
|---|---|---|
| Density ice | $\rho_{ice}$ | 910 kg m$^{-3}$ |
| Thermal conductivity ice | $K_{ice}$ | 2.10 Wm$^{-1}$K$^{-1}$ |
| Thermal conductivity rock | $K_r$ | 3.0 Wm$^{-1}$K$^{-1}$ |
| Heat of fusion ice | $H_{ice}$ | $334 \cdot 10^3$ J kg$^{-1}$ |
| Pressure-melting constant | $C$ | $7.4 \cdot 10^{-8}$ K Pa$^{-1}$ |

| Variable | Symbol | Unit |
|---|---|---|
| Overall shear stress near base | $\tau$ | Pa |
| Normal stress on stoss side (horizontal) | $\sigma_{stoss}$ | Pa |
| Normal stress on base (vertical) | $\sigma_v$ | Pa |
| Heat flow through obstacle | $Q_{ob}$ | W m$^{-2}$ |
| Heat production by frictional heating | $Q_{fr}$ | W m$^{-2}$ |
| Melt rate | $M_r$ | m s$^{-1}$ or mm yr$^{-1}$ |
| Pure sliding velocity | $V_{sl}$ | m s$^{-1}$ or m yr$^{-1}$ |
| Velocity component by pressure melting | $V_{pm}$ | m s$^{-1}$ or m yr$^{-1}$ |
| Velocity component by enhanced creep | $V_{cr}$ | m s$^{-1}$ or m yr$^{-1}$ |
| Water flow across CTB | $F_w$ | kg s$^{-1}$ m$^{-2}$ |

| Parameter | Symbol | Value |
|---|---|---|
| Ice thickness | $h_{ice}$ | 800 m |
| Surface slope | $\alpha$ | 1° |
| Height obstacle [1] | $h$ | 1 m |
| Width obstacle [1] | $w$ | 1 m |
| Spacing obstacles [1] | $\lambda$ | 4m |
| Geothermal heat flow [2] | $Q_{geo}$ | 0.05 W m$^{-2}$ |

[1] same as Weertman (1957)

[2] typical range: 0.03 – 0.07 W m$^{-2}$ or 30 - 70 mW m$^{-2}$

**Table 1**. Constants, variables and parameters with chosen values.

**Appendix**

To maintain a temperate layer below cold ice with a steep thermal gradient requires a heat flow through the cold-temperate boundary (CTB) of:

(A1)     $Q_{CTB} = K_{ice}(dT/dz)$  in $[\mathrm{W\ m^{-2}}]$

where $K_{ice}$ is the thermal conductivity of ice and $dT/dz$ the thermal gradient just above the CTB. In the case of borehole D near Jakoshavn Isbrae, the thermal gradient is 0.05 °C m$^{-1}$ (Lüthi et al., 2002), requiring $Q_{CTB}$ to be $2.1 * 0.05 = 0.105$ W m$^{-2}$, about twice the normal geothermal heat flow (see Fig. 2).

The transport of energy by water that passes through the temperate layer $Q_{LAT}$ and freezes above the CTB is given by:

(A2)     $Q_{LAT} = F_w\,H_{ice}$  in $[\mathrm{W\ m^{-2}}]$

where $F_w$ is the mass flux of water across the CTB, and $H_{ice}$ the heat of fusion of ice. A temperate layer will maintain its thickness if $Q_{LAT} = Q_{CTB}$, and will thicken if $Q_{LAT} > Q_{CTB}$. The required mass flux of water thus becomes:

(A3)     $F_w = K_{ice}(dT/dz)/H_{ice}$  in $[\mathrm{kg\ m^{-2}\ s^{-1}}]$

In the example of Borehole D, the required mass flux $F_w = 6.6 \cdot 10^{-7}$ kg m$^{-2}$ s$^{-1}$. Expressed as a melt rate, using equation (2), $M_r = 7.29 \cdot 10^{-7}$ m s$^{-1}$ or 23 mm yr$^{-1}$.

The pressure melting temperature is given by:

(A4)     $\Delta T_m = -C\Delta P$  in $[°\mathrm{C}]$

The overall pressure is:

(A5)     $P_{ice} = \rho_{ice}gh$

With $h_{ice} = 800$ m, it follows that $P_{ice} = 7.13 \cdot 10^3$ kPa, so that $\Delta T_m = -0.52$ °C.

Shear stress is given by:

(A6)     $\tau = \rho g h \sin \alpha$

With surface slope $\alpha = 1°$ and ice thickness $h_{ice} = 800$ m, the shear stress $\tau = 124$ kPa.

The concentration of horizontal normal stress acting onto a vertical stoss side is given by Weertman (1957), see equation (3). Using the same parameters as Weertman (1957): $h = 1$m, $w = 1$ m, $\lambda = 4$ m and the overall shear stress from equation (A6), $\tau = 124$ kPa, it follows that: $\sigma_{stoss} = 330$ kPa.

The pressure melting point at the stoss side is given by:

(A7)     $\Delta T_{stoss} = -C\sigma_{stoss}^n$     in $[°\mathrm{C}]$

Taking the value of $\sigma^n_{stoss}$ from equation (3) it follows that: $\Delta T_{stoss} = -0.024$ °C. This is the pressure melting point depression below ambient $T_m$, so that the $T_{stoss} = -0.544$ °C. Heat flow $Q_{ob}$ through an obstacle is given by Weertman (1957), see equation (7). For an obstacle 1 m long, and taking the value of $\Delta T_{stoss}$ from (A7), $Q_{ob} = 0.15$ W m$^{-2}$. For an obstacle 4 m long, but with all other parameters the same, the heat flow becomes: $Q_{ob} = 0.038$ W m$^{-2}$.