# Peer review of "Sliding of temperate basal ice on a rough, hard bed"

_The Cryosphere, 2016_

## Short Comment (SC1) · 6 Apr 2016

This is an interesting paper that draws attention to a possibly highly significant process. I shall not comment on the analysis, but would like to highlight evidence for grain boundary pressure melting beneath Svalbard surging glaciers. Ice formed during surges is well exposed at calving cliffs and in 'ice caves' (englacial and subglacial conduits), and has been the subject of recent detailed studies (see Lovell et al. 2015. Debris entrainment and landform genesis during tidewater glacier surges. Journal of Geophysical Research DOI: 10.1002/2015JF003509). A distinctive ice facies is commonly present at or close to the glacier beds, consisting of glassy, bubble-free ice, sometimes containing dispersed clots of silt. In some cases, filaments and ribbons of

bubbles are present, delineating crystal boundaries and apparently recording migration of gas between crystals. We interpret the facies as formerly wet, highly strained basal ice, in which pervasive grain boundary melting has occurred as a consequence of strain heating - the same process discussed by Krabbendam.

In Svalbard, this facies is typically < 1 m thick, occurring either as a single basal ice layer, or multiple thinner lenses between layers of sheared, debris-rich ice. The thickest basal layer of glassy ice we have observed was at Rabotsbreen, where it attains 10 m in places.

Bubble-free ice also occurs at the base of temperate glaciers, where it too appears to record the expulsion of gases along interconnected grain boundaries. At some locations (including Icelandic glaciers and at the base of Engabreen, Norway) one finds curious ellipsoidal water-filled lenses aligned parallel to flow, which may relect the concentration of excess water.

Ice in which net melting occurs contrasts with the classic conception of 'temperate ice' in which net meltwater production is zero or very small. In essence, excess grain boundary melting reflects an increase in enthalpy, mirroring the loss of potential energy during ice flow. For ice at the pressure-melting point, the key issue is whether this enthalpy can be evacuated from the system at a rate commensurate with its production.

---

## Referee Comment (RC1) · M. Montagnat (Referee) · 28 Apr 2016

This paper focuses on the basal ice motion in ice sheet conditions, by re-analysing the classical Weertman sliding model, and by suggesting new processes to take into account in order to better model and understand the physical mechanisms. In particular, the paper concentrates on the role of frictional heating at the base of ice sheets and glaciers, and on the impact of a specific rheology of temperate ice compared to cold ice.

Considering my skills, my comments will concentrate on what concerns the rheological model suggested for taking into account the role of temperate ice. The approach suggested to take into account frictional heating as a way to modify the amount of

temperate ice that can be encountered at the base of glaciers and ice sheets sounds very coherent with previous works on frictional heating in ice (and other materials), as far as I know. The author insists therefore on the fact that the ice could be, at least in some areas, mostly temperate, and that an appropriate rheology must be considered. Temperate ice is indeed known to deform at much faster rates than does cold ice. Several studies where made to show that, some are cited in this work (Morgan 1991), and some could be much deeper studied and used in the same direction, such as De La Chapelle et al. 1999 (GRL), and 1995 (Scripta Mat.), in order to infer the basic mechanisms that could explain the strain-rate increase in temperate ice. Temperate ice can be considered as ice with a liquid intergranular phase. It is therefore very similar to some geological materials, and some metals deformed at high temperature. Although classical rheological mechanisms such a dislocation creep are strongly enhanced at temperature very close (or at) the melting point, grain boundary sliding (GBS) could also be very efficient in such conditions. I therefore do not understand why this mechanism is not discussed in the present paper? De La Chapelle et al. (1995 and 1999) show that, when deforming ice with a liquid intergranular phase, they obtain a mechanical response with a very similar power law that in cold ice, except that the minimum strain rate reached can be more that 6 times higher (similar to what Morgan 1991 found). Nevertheless, their curves keep showing a power-law exponent of 3 for stress higher than $\sim 0.4$MPa, and close to 2 for lower values of stress. The value of the transition stress depends on the water content. These observations are therefore coherent with the fact that dislocation creep keeps dominating the deformation behavior of ice with an intergranular liquid phase, and that GBS, that should occur, is not dominating (would induce a stress exponent of 2, whatever the level of stress, and a grain size dependance, that was not tested in De La Chapelle's work), nor should dominate diffusional or solution-deposition creep (and therefore pressure-solution creep as evoked here?). The increase in strain rate is attributed to the fact that the liquid phase decreases the internal stress field (coming from strong strain heterogeneities between grains), and facilitates basal dislocation glide, and therefore deformation. Grain boundary migration could also be facilitated in such a condition (or at T close to Tm), and would also relax the stress heterogeneities, and facilitates deformation (such as during dynamic recrystallization).

Instead of using such observations (although coming from saline ice, and not "exactly" temperate ice), the author evokes a mechanism called "grain boundary pressure melting". This mechanisms, which is supposed to be supported by the work of Wilson et al. 1996, assumes that some grain boundaries (GB) would be submitted to higher stress, depending on their orientation compared to the maximum deviatoric stress. Such an assumption would not take into account the strong stress redistribution in polycrystalline ice, that lead to strong strain heterogeneities, with very weak relation between the level of deformation and grain orientation (Grennerat et al. 2012, acta Mater, for instance). And, indeed, Wilson et al. 1996 results tend to show these strong heterogeneities of the strain distribution, and therefore of melt at GB. To sum up, taking into account the fact that temperate ice rheology can be different than that of cold ice, especially the fact that activation energy and strain rates are much higher, is very important in the situation described in this paper. The relation between frictional heating and temperate ice is also well described and makes a lot of sense to me. I just do not see the interest of trying to find "another type" of rheology for temperate ice, especially since the one suggested here (GBPS) is very unlikely, in order to support the main assumption made in this work. Although few experiments exist on temperate ice, the work of Morgan, 1991, and De La Chapelle et al. (1995, 1998) can be efficiently used to enhance the main assumption.

Some specific comments:

- p 9, 2d paragraph: I don't think that results on temperate ice show that power-law creep does not adequately describe ice creep above -0.2°C. With increase in temperature, the activation energy can change, and therefore, we do not expect a linear relation between strain rate and temperature. To get rid of a power-law creep relation, one needs to plot minimum strain-rate as a function of stress... and show that this is

not linear. Which is not provided in Morgan 1991. Results by De La Chapelle et al tend to show that power-law creep remains even when a liquid phase exist at GB, which is mostly what happens when ice is temperate (Wilson et al. 1996)?

- p9, last paragraph: to my point of view, there are not enough information to be able to suggest a mechanism such as the one suggested here (GBPS), that has never been observed in ice, and that appears more than unlikely regarding knowledge about ice deformation behavior, with or without liquid layers... A discussion would nevertheless be required concerning the possibility of grain boundary sliding at such high temperatures.

- p10, 1st paragraph: As far as I know, pressure solution requires different phases to be present, in order for some to be dissolved under local pressure, and migrate is some fractures together with the fluid phase, and re-precipitate further away (see J-P. Gratier, D. K. Dysthe and F. Renard. The role of pressure solution creep in the ductility of the Earth's upper crust. Advances in Geophysics, vol. 54, 2013). I do not see at all how this can occur in the temperate ice layer at the bottom of glaciers and ice sheets...

- p11, 1st paragraph: Once again, there are not enough proof or information to assess so directly the occurrence of some grain boundary pressure melting... Power-law creep can also be fast, if accommodation mechanisms are efficient (dynamic recrystallization, GBS, liquid intergranular phase)... so it can not be ruled out so easily.

- p12, 1sr paragraph: same comment about power-law creep being ruled out...

- Conclusions: To my point of view, they need to be rewritten by being much more precise about the terms used. What is basal creep?? is it creep due to basal dislocations glide? Creep at the base of the glaciers? Therefore, it can be 10 times faster only because the presence of a liquid layer and high temperature enhancing dislocation creep... Dynamic recrystallization is evoked briefly, without being much explained before. It needs a lot of dislocations to be activated, and therefore is in favour of dislocation creep... It is more efficient at high temperature. Its occurrence should however

be reduced by the presence of a liquid phase at GB, since this liquid phase will most probably be very efficient to relax the local stress field, and absorb dislocations in excess, that are responsible for dynamic recrystallization mechanisms (nucleation and grain boundary migration). GBS would be much more likely to occur than GBPM in order to be associated with dislocation creep.

As a summary, it seems quite important that the hypotheses performed here about temperate ice rheology be clarified, and cleaned of non or too lightly justified hypotheses, that, as far as I understood, are not strictly required to provide the main message of this work.

Sincerely

M. Montagnat April 28th

---

## Referee Comment (RC2) · D. Cohen (Referee) · 29 Apr 2016

This is a very interesting paper on a topic of great importance: sliding is a dominant form of ice motion in many parts of glaciers and ice sheets but its parameterization remains poorly constrained which leads to lots of uncertainties in prognostic models, probably the largest uncertainty in glacier flow models. Having just read the review by Maurine Montagnat I will not comment on the rheology of basal ice. All was said there.

First let me state that I feel this article is extremely relevant and important because sliding continues to be understudied in glaciology despite everyone agreeing that it is a major source of uncertainty. Linked to sliding, there are also clear evidences that basal ice is physically different from bulk ice, displaying different mechanical behavior. Yet this

difference is not included in numerical models of glaciers and ice sheets. The problem with sliding, I think, is two-fold: (1) there is a lack of data making it hard to have a comprehensive view of sliding processes and their relative importance. This also makes it difficult to build a coherent verifiable mathematical model; (2) ice-sheet modelers usually perform diagnostic models of present-day ice sheets/glaciers (what the author calls "near steady-state situations") using inverse models to obtain the slip coefficient (or the "friction parameter") of the sliding law neglecting the physics of sliding and the dependence of that coefficient on basal conditions (water, debris, basal topography, etc). Attention has thus shifted to a fitting problem rather than to the physics. I fully agree with the author that inverse models are not appropriate for long-term diagnostic simulations and for palaeo-glacier modeling. To perform such simulations, one needs a better understanding of basal processes and a better mathematical description of sliding based on physics.

The paper is constructed around the sliding theory of Weertman of 1957 arguing that Weerman's model of sliding is invalid because it wrongly scales pressure melting with obstacle length instead of obstacle height, and it uses power-law creep with an exponent of 3 instead of grain boundary sliding with a smaller exponent and with much higher strain rates. The author then argues that his modified conceptual model of sliding could explain the fast sliding observed in parts of bedrock-dominated ice sheets (order 100 m/year). First I am not convinced by this approach which tries to deconstruct Weertman's theory. Lots of new data have been acquired since 1957 and one cannot blame Weerman's for not knowing them. Weertman was a precursor of sliding theory and his first model clearly represents an extreme simplification of the process. Yet it captured some very essential components. In my opinion some parts of the text come out too strongly as a critic of Weertman's work. Second, I am also not convinced by one of the main point of the discussion, that pressure melting, even an enhanced form, has any significant impact on ice-sheet sliding. Weertman's calculations indicate a speed of about 0.01 m/year or less for pressure melting for a 1 meter obstacle. Even using an enhancement factor of 100 this mode of sliding is negligible (see below).

Rheology, as discussed in the paper, can have a much greater impact on sliding and this is clear from the data. Also as discussed in the paper, the basal thermal regime and the temperate ice layer, including the effects of basal meltwater and friction, water fluxes, surface water influx, are key elements to properly quantify sliding, and, as correctly pointed out, are not often taken into account. I find Figure 6 very useful as a conceptual model of what is going on at the bed.

Regarding the pressure melting mechanism, using the author's numbers in Section 2 and the appendix, I come up with a pressure-melting velocity Vpm, Equation 4, of 1.5 cm/year (Equation 4 should read as Q/(rho H)). This is completely negligible when talking about ice sheets that move ca 100 m/year. As stated by Weertman in 1957, pressure melting is unimportant for "large" bumps. One meter here is already a large bump. Given these numbers, I am not sure of the relevance of pressure melting for sliding. Whether the scaling is length or height (Section 5) the numbers will remain extremely small. I agree, however, that height is probably a better scaling than length.

Again in Section 6 I am not sure how pressure melting can explain the increase in sliding due to surface melt water input given its negligible effect (cm per year). Even increasing stoss side pressure by a factor of 10 would not help explaining the high sliding speeds observed underneath ice streams. I think surface water is an important issue and has an impact on sliding. I just don't see how the arguments presented here for enhanced pressure melting can explain the observed effects. May be I missed something in which case quantifying this effect in this section would be helpful. The same argument goes for Section 8.1 regarding the rate-controlling mechanisms: numbers are necessary to illustrate the importance of stoss side pressure melting. In my opinion, for bumps of the size of 1 m or greater, this has no effect. Other thermal effects discussed in the paper, like debris-bed friction, basal melt, viscous heating, surface meltwater influx and drainage, are more important and control, in some way, the sliding speed and the temperate ice layer.

Overall I think the paper brings a new and interesting point of view of glacier sliding. It
clearly explains what the problems are with sliding as it is used in glacier models today. The paper is overly focused on Weertman's theory and it lacks first-order estimation of some of the basic controlling mechanisms.

Specific comments

- Page 1 last line: Probably "near steady-state situations" should be replaced with something like "near-instantaneous situations". Clearly most of the ice sheets/glaciers where this inverse model is being used are not in a steady state.

- Page 2 Line 10. I think the author meant to refer to Cohen et al 2005 and not Cohen et al 2006. That reference is actually missing from the bibliography. Cohen et al 2000 in the bibliography is not cited in the text. The 2005 reference is: Cohen, D., N. R. Iverson, T. S. Hooyer, U. H. Fischer, M. Jackson, and P. L. Moore (2005), Debris-bed friction of hard-bedded glaciers, J. Geophys. Res., 110, F02007, doi:10.1029/2004JF000228.

- Page 2 Line 24. I would insert the word "either" before "pose problems"

- Page 3 Line 20. I think the mechanisms of sliding are clear: ice at the melting temperature contains water and water between ice and bedrock forms a thin lubricating layer with near zero shear resistance that allows ice in contact with the bedrock to have non-zero velocities. The question is how to quantify sliding and what glaciological parameters control it.

- Page 4 Line 20 Equation 4. There is an typo/error. It should be: $V_{pm} = Q_{obs}/(H_{ice} * \rho_{ice})$.

- Page 4 Line 24. Strictly speaking in regelation ice does not flow around the obstacle. That's the viscous part of motion. In regelation ice melts on one side, the water flows to the other side and refreezes there. May be change wording.

-Page 5 Line 2. The vertical stress could even be higher than the effective pressure since, due to melting, there is a component of ice flow towards the bed that creates a vertical downward force on the debris. This force could be significant and further

increase basal friction. See Hallet 1979, 1981, and Cohen et al 2005. Hallet, B. (1979), A theoretical model of glacial abrasion, J. Glaciol., 23, 39 – 50. Hallet, B. (1981), Glacial abrasion and sliding: Their dependence on the debris concentration in basal ice, Ann. Glaciol., 2, 23 – 28.

- Page 6 Line 11 (ii). Strictly speaking this is not true. There will be differences in temperature in temperate ice due to differences in stresses (if only with depth). These temperature differences will cause thermal gradients and heat fluxes (arguably small). These gradients will only serve to melt ice or freeze water. See Lliboutry 1993.

- Page 7 Line 10. The use of the words 'cold patch' is confusing. The ice is at the melting temperature so it's not cold. I think the term cold patch should be restricted to cold ice not ice at the melting temperature that is colder because under a higher pressure. See also Figure 1.

---

## Short Comment (SC2) · 2 May 2016

In addition to the three comments & reviews already provided, I would like to point out that the manuscript in its present form should consider the relevance of the present discussion and inferences of basal properties in light of the most recent field measurements. Christianson et al. employed various geophysical techniques to constrain the subglacial strata. They conclude that these are made up of high-porosity water-saturated till, which lubricates the ice stream. As this is in contradiction to the hard-bed assumption for NEGIS, as employed in the manuscript, I suggest a thorough discussion of the different end-member cases.

With the now starting EastGRIP project (http://eastgrip.org/), at least for one part of

the ice stream, we might know in a couple of years which basal properties do in fact prevail.

Knut Christianson, Leo E. Peters, Richard B. Alley, Sridhar Anandakrishnan, Robert W. Jacobel, Kiya L. Riverman, Atsuhiro Muto, Benjamin A. Keisling, Dilatant till facilitates ice-stream flow in northeast Greenland, Earth and Planetary Science Letters, Volume 401, 1 September 2014, Pages 57-69, ISSN 0012-821X, http://dx.doi.org/10.1016/j.epsl.2014.05.060

---

## Author Comment (AC1) · 5 May 2016

Reply to interactive comment by Doug Benn: 'Ice facies evidence of grain boundary melting' on "Basal sliding of temperate basal ice on a rough, hard bed: pressure melting, creep mechanisms and implications for ice streaming" by M. Krabbendam

Thank you for pointing out the paper by Lovell et al. (2015); it is really interesting, and I shall, in the revised version, indeed discuss their paper. Indeed, the formation of the basal 'dispersed facies' of bubble-free glassy ice by strain induced metamorphism, including partial melting, is pretty much the process I had in mind within the temperate ice that I pursue in the manuscript, and it is good to see this being documented, even though the glaciological situation (surging glacier vs. ice sheet ice streaming) is

somewhat different.

However, we need to be careful to distinguish 'basal ice facies' from 'temperate ice'. Firstly, it is of course possible to have temperate ice that is not a basal ice facies, see for instance the Glacier de Tsanfleuron (Tison and Hubbard, 2000), where there's clearly different ice facies, but the glacier appears to be temperate throughout. It would be interesting to study the different mechanical behaviours of these different ice facies all at the pressure melting point, to study the effects of grain size and bubble/debris content. Secondly, even if the 'bubble-free' ice is produced at or near the base by partial melting, it is possible to cool this ice subsequently to below the melting temperature. This is potentially the case in large polythermal ice sheets (e.g. Greenland) where large scale folding (with 100s metres amplitude, for instance: Bell et al. 2014) have brought up 'featureless' basal ice higher up into colder ice. Given sufficient time, thermal conduction will cool these uplifted basal ice facies to well below the melting temperature. So 'basal ice facies' and 'temperate ice' do not necessarily equate: one is an ice facies, the other is purely a thermal state.

References:

Bell, R. E., Tinto, K., Das, I., Wolovick, M., Chu, W., Creyts, T. T., Frearson, N., Abdi, A., et al.: Deformation, warming and softening of Greenland's ice by refreezing meltwater, Nature Geoscience, 7, 497-502, doi:10.1038/ngeo2179, 2014.

Lovell, H., Fleming, E. J., Benn, D. I., Hubbard, B., Lukas, S., Rea, B. R., Noormets, R., and Flink, A. E: Debris entrainment and landform genesis during tidewater glacier surges., Journal of Geophysical Research: Earth Surface, 120, 1574-95, doi: 10.1002/2015JF003509, 2015.

Tison, J. L. and Hubbard, B.: Ice crystallographic evolution at a temperate glacier: Glacier de Tsanfleuron, Switzerland, Geological Society, London, Special Publications, 176, doi: 10.1144/GSL.SP.2000.176.01.03, 2000.

---

## Author Comment (AC2) · 5 May 2016

Reply to interactive comment by Olaf Eisen 'Update of literature and discussion needed'

Thank you for pointing out the paper by Christianson et al. (2014); it is really interesting, and I shall, in the revised version, indeed discuss their paper. Christianson et al. (2014) note that the bright radar reflectivity points to a thawed bed with significant water flow beneath the main trunk of the NEGIS, and that the bed is rough, so there is agreement on these issues.

The problem with many subsurface geophysical methods is that they generally do not

provide unique solutions. The seismic properties of temperate ice, say with a water content of 1-3%, were not discussed by Christianson et al. (2014), so the reader cannot judge whether the lower P-wave and S-wave velocities (which constitute the actual observations) are unique indicators for dilatant till, or that they may be explained by some other material. Could many of the geophysical observations by Christianson et al. (2014) not (also) be explained by the presence of a layer of temperate ice? Certainly Peters et al. (2012) show a very strong seismic attenuation (or: a low seismic Q) near the melting point, which suggests a much lower P-wave velocity in temperate ice.

A problem with 'deforming till' is that it will continually move downstream with the ice, so it needs to be replenished by erosion upstream in order to sustain ice streaming, if deforming till was the main rate-controlling factor. Christianson et al. (2014) suggest as potential sources:

(A) 'geothermally altered rocks', related to the area of high geothermal heat flux, invoked by Fahnestock et al. (2001) to explain the very high basal melting rates they deduced. As discussed in the Manuscript, Fahnestock et al. (2001) did not take frictional heating into account, so the existence of such a geothermal high needs to be questioned. Apart from this, what rocks are we talking about here, and why would they produce more till?

(B) Reworking of ice-marginal deposits left over from a previous glaciation. This begs the question why such deposits were so concentrated in one place. Furthermore, typically Pleistocene deglaciated crystalline basement areas (e.g. Laurentide and Fennoscandian ice sheets) have experienced similar glaciation/deglaciation cycles, and thus could have similar ice-marginal deposits left from previous glaciations. However, where warm-based ice conditions are known to have prevailed, till is absent or patchy at best (see Manuscript for references).

Furthermore, given the observed roughness, with bedrock humps several 10s of me-
tres high (Christianson et al. 2014), it should be noted that a deformable till will as such not help in moving ice over/around bedrock obstacles, unless the till covers these obstacles, which they do not seem to do. In short, the presence of a deformable till as a rate controlling factor below the NEGIS is by no means proven. Instead, the high melting rates deduced by Fahnestock et al. (2001) are very likely to lead to the production of a basal temperate layer of significant thickness.

Having said all this, it is of course possible that pockets and smears of till do exist below the NEGIS and that this is overlain by a temperate ice layer – deformable till and basal temperate ice are not mutually exclusive. The main purpose of discussing the NEGIS in the Discussion is to try to make other researchers think about, and check the possibility of, the potential existence of a basal temperate ice layer below modern and ancient ice streams.

References:

Christianson, K., Peters, L. E., Alley, R. B., Anandakrishnan, S., Jacobel, R. W., Riverman, K. L., Muto, A., and Keisling, B. A.: Dilatant till facilitates ice-stream flow in northeast Greenland, Earth and Planetary Science Letters, 401, 57-69, doi: 10.1016/j.epsl.2014.05.060, 2014.

Fahnestock, M., Abdalati, W., Joughin, I., Brozena, J., and Gogineni, P.: High geothermal heat flow, basal melt, and the origin of rapid ice flow in central Greenland., Science, 297, 2338-42, doi: 10.1126/science.1065370, 2001.

Peters, L. E., Anandakrishnan, S., Alley, R. B., and Voigt, D. E.: Seismic attenuation in glacial ice: A proxy for englacial temperature, Journal of Geophysical Research: Earth Surface, 117, F02008, doi:10.1029/2011JF002201, 2012.

---

## Author Comment (AC3) · 5 May 2016

Reply to interactive comment by M Montagnat (referee)

Thank you very much for a very constructive and helpful review. I will make changes in the revised manuscript according to the suggestions, but a few points are worth discussing beforehand, if only to other interested readers.

The experiments of de La Chapelle et al. (1995; 1999) are, unfortunately, not strictly relevant for deformation of temperate ice. These experiments were performed with pure ice at -5 °C and -13°C, with the intragranular liquid provided by a brine, with a much lower melting point. Thus, this experiment contained two materials with different melting temperatures, where the liquid and the solid cannot freely interact through normal melting/freezing. Thus, whereas these experiments are interesting to study dislocation creep in the present of a different liquid phase (compare the hydrolytic weakening of geological materials, e.g. the difference in behaviour of 'dry quartz' and 'wet quartz'), they are in essence experiments on cold ice. Therefore, it is not surprising that these experiments document a n∼3 power-law exponent, and that dislocation creep remains dominant. But, the results cannot be extrapolated to true temperate ice, where ice and water are in thermodynamic equilibrium, which can melt or refreeze at the smallest perturbation of temperature or stress.

The behaviour of temperate ice is NOT very similar to other geological materials and metals at high temperatures. No other such material (with the apparent exception of plutonium!) shows the near-unique behaviour of $H_2O$, namely that the liquid phase is denser than the solid phase. This fundamentally different phase-transition behaviour is likely to lead to a fundamentally different deformation behaviour. In other words, the whole concept of pressure melting, namely that higher pressure leads to a lowering of the melting temperature only works for $H_2O$. I will emphasise this difference more strongly in the revised paper as follows: "Comparisons with other materials (rocks, metals) are no help, as water is (almost) unique in that its solid phase is less dense that its liquid phase. This fundamentally different phase transition behaviour means that the liquid-solid mixture of $H_2O$ may have a fundamentally different deformation behaviour at the pressure melting point. "

Both the dramatic and sudden increase in strain rate close to the melting temperature, and the (albeit limited) evidence for near-Newtonian (n ∼ 1) behaviour in temperate ice do suggest there is a switch in rate-controlling deformation mechanism, rather than 'merely' a variation of dislocation creep.

Grain boundary sliding (GBS) is indeed commonly invoked to explain sudden weakening in a variety of materials, i.e. by superplasticity. This is worth emphasising and discussing, and I will do so in the revised paper. GBS is thought to be favoured by

small grain size, and (for most grain shapes) needs to have an accommodation mechanism (for grain edges to move past each other) which may be diffusion-dominated or dislocation-creep dominated. GBS behaviour has been simulated in ice in experiments at temperatures between -37°C and -80°C and with grain sizes between 3-40$\mu$m (Goldsby and Kohlstedt, 1997, 2001; Goldsby and Swainson, 2005). Such conditions may be relevant for icy planets, but are very different from the temperate ice below terrestrial ice sheets under consideration here. Basal temperate ice is observed to be coarse (mm-cm scale grain size, e.g. Tison and Hubbard (2000), 3-4 orders of magnitude coarser than in the experiments mentioned above, again not compatible with the GBS experiments. GBS in rocks can destroy, or prevent the formation of, crystal C-axis fabrics (e.g Krabbendam et al. 2003); whether this occurs in ice or not is ambiguous (Goldsby and Kohlstedt, 2001; 2002; Duval and Montagnat, 2003; Goldsby and Swainson, 2005). Overall, I feel that GBS is unlikely to be the rate-controlling factor in the deformation of coarse, temperate ice, but I agree this needs to be discussed in the revised Manuscript and I will do so.

I will reword the part on 'grain boundary pressure melting', but this is not "another" new mechanism: it has been documented before by Barnes and Tabor (1966, 1967) and Barnes et al. (1971), but referred to as 'internal pressure melting' or 'grain boundary melting'. I will change the manuscript to reflect this better. Extensive melting along grain boundaries was observed by Wilson et al. (1996) whereas and Hubbard et al. (2000) and Lovell et al. (2015) also invoke melting and refreezing to explain the deformation metamorphism in basal temperate ice into bubble-free basal ice. There is thus ample evidence that melting along grain boundaries does occur in deforming temperate ice. In terms of deformation mechanisms, can 'grain boundary melting' be seen as a very fast version of grain boundary diffusion creep (Coble Creep)? This is hinted at by Goldsby & Kohlstedt (2001). This is then likely to result in n<3 behaviour, although the problem of grain-size sensitivity remains. For some reason, the process has been rather ignored in ice rheology – maybe because it is unlikely to operate in other materials?

I think we all agree that too little is known about the mechanical behaviour of temperate ice. . ..

References:

Barnes, P. and Tabor, D.: Plastic Flow and Pressure Melting in the Deformation of Ice I, Nature, 210, 878-82 doi:10.1038/210878a0, 1966.

—: Plastic Flow and Pressure Melting in the Deformation of Ice I, IAHS Symposium, 303-15, 1967.

Barnes, P., Tabor, D., and Walker, J. C. F.: The friction and creep of polycrystalline ice, Proceedings of the Royal Society of London. A. Mathematical and Physical Sciences, 324, 127-55, doi: 10.1098/rspa.1971.0132, 1971.

De La Chapelle, S., Duval, P., and Baudelet, B.: Compressive creep of polycrystalline ice containing a liquid phase. , 33(3), . Scripta metallurgica et materialia, 33, 447-50, doi:10.1016/0956-716X(95)00207-C, 1995.

De La Chapelle, S., Milsch, H., Castelnau, O., and Duval, P.: Compressive creep of ice containing a liquid intergranular phase: Rate-controlling processes in the dislocation creep regime, Geophysical Research Letters, 26, 251-54, doi: 10.1029/1998GL900289, 1999.

Duval, P. and Montagnat, M: Comment on "Superplastic deformation of ice: Experimental observations" by DL Goldsby and DL Kohlstedt, Journal of Geophysical Research: Solid Earth, 107, Pages ECV 4-1–ECV 4-2, doi:10.1029/2000JB000336, 2002.

Goldsby, D. L. and Kohlstedt, D. L.: Grain boundary sliding in fine-grained ice I, Scripta Materialia, 37, 1399-406, 1997.

—: Superplastic deformation of ice: Experimental observations, Journal of Geophysical Research, 106, 11017-30, doi: 10.1029/2000JB900336, 2001.

—: Reply to comment by P. Duval and M. Montagnat on "Superplastic deformation of

ice: Experimental observations", Journal of Geophysical Research: Solid Earth, 107, ECV 17-1–ECV 17-5, doi:10.1029/2001JB000946, 2002.

Goldsby, D L and Swainson, I: Development of C-axis fabrics during superplastic flow of ice., Eos, Transactions, American Geophysical Union., 86, C51B-029, 2005.

Hubbard, B., Tison, J. L., Janssens, L., and Spiro, B. : Ice-core evidence of the thickness and character of clear-facies basal ice: Glacier de Tsanfleuron, Switzerland, Journal of Glaciology, 46, 140-50, doi: 10.3189/172756500781833250, 2000.

Krabbendam, M., Urai, J.L., and van Vliet, L.J.: Grainsize stabilization by dispersed graphite in a high-grade mylonite: an example from Naxos (Greece), Journal of Structural Geology, 25, 855-66, 2003.

Lovell, H., Fleming, E. J., Benn, D. I., Hubbard, B., Lukas, S., Rea, B. R., Noormets, R., and Flink, A. E: Debris entrainment and landform genesis during tidewater glacier surges., Journal of Geophysical Research: Earth Surface, 120, 1574-95, doi: 10.1002/2015JF003509, 2015.

Tison, J. L. and Hubbard, B.: Ice crystallographic evolution at a temperate glacier: Glacier de Tsanfleuron, Switzerland, Geological Society, London, Special Publications, 176, doi: 10.1144/GSL.SP.2000.176.01.03, 2000.

Wilson, C. J. L., Zhang, Y., and Stüwe, K.: The effects of localized deformation on melting processes in ice, Cold Regions Science and Technology, 24, 177-89, doi:10.1016/0165-232X(95)00024-6, 1996.

---

## Referee Comment (RC3) · M. Montagnat (Referee) · 9 May 2016

Thank you for this well documented answer. I am looking forward to read the new version of the paper, and I am sure that your suggestions concerning rheology of temperate ice will enable to create some interest in trying to better understand this rheology, and over all, the impact of such a rheology on ice sheet flow modeling.

Sincerely Maurine

---

## Author Comment (AC4) · 17 May 2016

Reply to interactive comment by D Cohen (referee) on "Basal sliding of temperate basal ice on a rough, hard bed: pressure melting, creep mechanisms and implications for ice streaming" by M. Krabbendam

Thank you very much for a very constructive and helpful review, this will help me to provide a better paper.

It was not my intention at all to deconstruct or criticise Weertman's original model; all I wanted to do is to point out it is not applicable to temperate ice with the original assumptions, which, unfortunately, are at times still perpetuated in probably inappropriate conditions. I will reword and rephrase, where appropriate, to be more respectful to Weertman's original model, which of course was pioneering, and has become classic.

I agree that, for larger-scale obstacles, creep is more important than pressure melting against the obstacles: in a sense the order in which I presented the manuscript probably had more to do with the order of my thinking, rather than the order in which the ideas are best presented. It is possibly better to start dealing with the creep component (which will become somewhat longer due to the comments of the other reviewer). So I've decided to re-arrange the manuscript as follows:

1 Introduction

2 Basal meltwater production by frictional sliding

3 Growing and maintaining a temperate ice layer

4 The creep component in temperate ice (this will be expanded to take care of M Montagnat's comments)

5 The pressure melting component

6 Stoss-side pressure melting in temperate ice

7 Effect of surface water input on temperate ice on a rough bed

8 Critical obstacle size

9 Discussion

10 Conclusions

The critical obstacle size will indeed need to be discussed, and I will make a comparison with the classic Weertman sliding model. In doing so, if one say assumes that pressure melting in temperate ice will be twice as fast for a given height of an obstacle, but the creep rate is say about 5x as fast, than the critical obstacle size will in fact be lower than in the classic Weertman model. Having said that, Kamb and LaChapelle

(1964) noted that whilst qualitatively Weertman's theoretical concept is correct, in nature and experiment their critical obstacle size is much larger (about 1 metre). Overall, for metre-scale obstacles, creep probably dominates, but pressure melting may well be crucial for debris, which normally is < 1m across.

As to the specific comments; these all make sense, and I will deal with these, mainly through rewording or rephrasing. Two deserve some reply:

"Page 5 Line 2. The vertical stress could even be higher than the effective pressure since, due to melting, there is a component of ice flow towards the bed that creates a vertical downward force on the debris. This force could be significant and further increase basal friction".

Yes, that is strictly true, but that concerns the very localised contact stress of a debris cobble onto the bed. On this contact the friction coefficient would also be much higher (rock on rock friction), probably in the order of mu = 0.5 or so. Since I use the low friction coefficient averaged over a large area, I have also used the vertical stress over a large area, rather than looking at the very localised stresses.

"Page 6 Line 11 (ii). Strictly speaking this is not true. There will be differences in temperature in temperate ice due to differences in stresses (if only with depth). These temperature differences will cause thermal gradients and heat fluxes (arguably small). These gradients will only serve to melt ice or freeze water. See Lliboutry 1993.".

Again, this is strictly true, but here I'm referring here to a bulk thermal gradient, capable of transporting significant heat from the base to the CTB. This bulk thermal gradient is zero (or rather it is slightly negative, due to the increase in cryostatic pressure). I will reword this, to make it clear.

Again, thank you very much for a thoughtful review

---

## Referee Comment (RC4) · Anonymous Referee #3 · 18 May 2016

I am sympathetic to the motivation for this paper: Weertman sliding is indeed flawed in important ways, and heat transferred by subglacial water flow and produced by debris-bed friction may affect sliding physics. More broadly, the problem of how glaciers slide rapidly over hard, rough beds is an important one. This paper, however, has serious deficiencies:

1) Most importantly, it contains little new analysis or data that help shed light on sliding physics. For example, the calculation of p. 4 yields conclusions that could have been reached without the calculation (see below, comments on p. 4, 1-21) and is used inappropriately to assert that the Weertman model is "illogical" (p. 4, 23-26). Too much of the paper consists of inferences not supported by data or relevant formal analysis.

2) Misconceptions/errors indicate a muddled understanding of relevant physics related to sliding (e.g., p. 7, 5-7; p. 7, 8-11; p. 8, 26; p. 10, 26-27; p. 11, 19) and ice rheology (e.g., p. 9, 8-15). The attempt to assess the extent to which temperate ice obeys a power-law flow rule by considering the dependence of strain rate on temperature, rather than on stress, is a particularly major error.

3) References are used inappropriately to support conclusions (p. 9, 8-15; p. 9, 8-23).

4) Inadequate justification is provided for some of the paper's assumptions (p. 5, 2; p.7, 5-7; p. 8, 1-4; p. 8, 5)

5) The introduction would benefit from substantial revision. The attempt to motivate the subject of this paper (i.e., sliding mechanics) with bullets 1-5 is not successful (see my comments below on p. 2-3).

Specific comments keyed to page and line numbers:

p. 1, 14-15. "Thermal equilibrium" is vague. There is "thermal equilibrium" in Weertman's model. The intended meaning needs to be clarified.

p. 1, 18. Power Law Creep should not be capitalized.

p. 2, 13-15. "The essence of Weertman's sliding model is that basal ice movement past an obstacle is controlled either by stoss-side pressure melting around the obstacle or by ductile flow enhanced by stress concentrations near the obstacle, whichever is the fastest." This wording is a bit misleading. It suggests that either pressure-melting or ductile flow occurs, when regardless of bump size, there will always be components of both, albeit with one more important than the other (except for the transition bump size for which they equally important).

p. 2. 25-29. It is not clear why fast ductile flow and soft ice somehow contradict Glen-type power law flow (i.e., the nonlinear dependence of strain rate on stress).

p. 2-3. 30-4. Although these are valid observations, the author needs to be more

explicit about how they contradict pressure melting or a power-law ice rheology.

p. 3 12-18. These comments on basal thermal regime and basal hydrology in Greenland come as a surprise because there is no allusion to Greenland earlier in the introduction. Such an allusion is necessary because temperate ice of thicknesses much larger than bump size and water access to the glacier bed are, of course, normal for temperature glaciers. If sliding OF THE GREENLAND ICE SHEET, is how the author wants to motivate this paper, then that should be made clearer at the beginning of the introduction.

p. 3. Omit "worked" before "example" here and elsewhere.

p. 4. 13. "was seen" indicates that this was actually observed. Express this differently here and where used elsewhere.

p. 4 "is then", earlier "was seen". Here and elsewhere the author tends to change tense in midstream.

p. 4. 1-21. The reason for this calculation is unclear. The points made after it– that cavity formation can impede heat transport and that regelation speed decreases linearly with bump length–could be made without presenting the calculation. Even if there were better motivation for presenting this calculation, the source of numerical values, such as those for ice thickness and bed shear stress, is unclear (not in the text or appendix but presumably from somewhere in Greenland) and the choice of bump size and spacing is seemingly arbitrary.

p. 4, 23-26. The author concludes at the end of the calculation: "This implies that ice flowing around an obstacle that is, say, four times longer than another obstacle (Fig. 1b), would be four times slower, even though this obstacle is more streamlined (having a longer aspect ratio). This result is illogical, contradicts most observed geomorphology (Stokes and Clark, 1999; Bradwell et al., 2008), and is a major weakness of the Weertman model." This decrease in speed with elongation is a major weakness of

the Weertman model ONLY if one neglects viscous flow in the Weertman model, as the author does here. And how can the author consider only pressure melting and refreezing–most relevant to bumps less than 0.5 m in wavelength–and assume that the calculation has relevance to the much larger landforms considered by geographers like Stokes and Clark? The Weertman model is indeed flawed, but this calculation adds nothing new to the subject, and the verbiage toward the end of the paragraph is misleading.

p. 5, 2. Explain why this approach is valid. In Hallet's abrasion model (1979; 1981) for example, debris-bed friction is independent of normal stress (and effective normal stress) and instead depends on the rate of ice convergence with the bed, so the assumption made here requires justification. Even for a flat bed, ice will converge with it due to basal melting, and that process exerts a downward drag on clasts, increasing friction between them and the bed. I think equation 5 can, in fact, be justified, but the necessary justification is not provided here.

p. 6. 11. A temperate ice layer, in fact, has a thermal gradient–one that reflects the decrease in melting temperature with pressure, as pictured in Figure 1 of this paper. Rewrite.

p. 6. This page-long digression ("Intermeezzo") distracts from the theme of the paper and will leave readers wondering what this paper is supposed to be about.

p. 6 34. This is a melt rate rather than a flux, which begs the question why melt rate was not expressed in Equation 6 with these units, as it normally is, either by defining the heat of fusion volumetrically or including density.

p. 7. 5-7. The assertion here that heat flow thorugh advection by flowing water is "more efficient" than heat flow due to thermal gradients in rock may be correct but is not demonstrated here or later in this paragraph. Also, if some of the frictional and geothermal heat is advected by water, why doesn't equation 6 reflect that? It should have a heat sink term in it associated with advection by flowing water.

p. 7. 8-11. The physics here is muddled. The ice temperature will be pinned everywhere along the bump surface at values set by the distribution of ice pressure (unless the temperature of the water in the film that divides ice from rock is not in equilibrium with the ice temperature, which seems unlikely). Although the water flow will cause some extra melting, if lee-side cavities do not form, the thermal gradient will be set by the pressure deviation from hydrostatic on both the stoss and lee sides of the bump. For a reason I don't understand, the author is assuming pressure is only important on the upstream side (see Figure 1c also). Also the word "cold patch", as used in the classic paper by Robin (1976) and by subsequent textbook writers (e.g. Hooke), describes ice below the pressure melting temperature (PMT). To use it in this context, where all ice is at the PMT, is thus confusing.

p. 8, 1-4. Debris-bed friction can be affected by water flux to the bed only if water can gain entry to zones where there may be small cavities beneath debris particles. This requires moving water from the channel at the point of entry to the bed out through the thin film that divides debris-bearing ice from bedrock. This propagation of pressure will be diffusive and slow, so it is unclear how much a sudden increase in warm meltwater will really decrease effective pressure and thereby reduce frictional drag.

p. 8, 5. The thermal gradient towards stoss surfaces depends not only on the temperature of the incoming warm water but on the distance between it and the stoss surfaces of bumps. Because the distances between channels carrying water and stoss surfaces is poorly known and could be far larger than the distance across a bedrock bump, the importance of this effect for stoss-side melting is uncertain, contrary to the certitude of the statement made here. Perhaps the author is assuming that all of the warm water in a Das-like event moves in the thin film that divides ice from rock. If so, that is a dubious assumption.

p. 8, 26. "Weertman (1957) assumed that the creep component of ice flowing around a hard obstacle worked with a rheology" according to 'Glen's Flow law', albeit enhanced by stress concentration on the stoss side." Again here, the author makes the error

of assuming deviatoric stresses are concentrated only on the stoss sides of bumps in Weertman's theory. Rather, deviatoric stresses are symmetric across bumps in the theory, with lee-side deviatoric stresses equal to but opposite in sign of those on stoss surfaces.

p. 8, 27. "strain-rate" rather than "strain".

p. 8, 28. Power Law should not be capitalized, here and elsewhere.

p. 9, 3. "comparisons" "suggests" Correct subject-verb correspondence.

p. 9, 9. In fig. 3 strain rate is plotted, not strain as reported here.

p. 9, 8-15. Here the author asserts that if the log of strain rate plots as a straight line against the reciprocal of temperature, then power law creep is indicated. He needs to be aware that power-law creep is defined on the basis of the relationship between stress and strain rate, rather than between stress and temperature. Note that Morgan (1991) (the source of the data reproduced here) never commented on whether his data conformed to power law creep rules because all of his tests were done at a single stress (0.1 MPa).

p. 9, 8-23. This point of the paragraph–that temperate ice obeys a near Newtonian flow rule–could conceivably be correct, given the relative lack of work on warm ice, but is not convincingly argued here. Other authors cited, such as Byers et al (2012) and Chandler et al (2008), did indeed suggest values of n near 1.0 but were careful to attribute those low values to low deviatoric stresses, for which there is some micromechanical justification for low n. For ice near glacier beds, however, and particularly near bumps, deviatoric stresses will tend to be high, and thus justification for low values of n is weak.

p. 9, 31. Why does the abbreviation, GBPS, not coincide to the first letters of "grain boundary pressure melting"? And why choose a new term here when there is ample discussion of this sort of deformation in the literature, much of which is not cited here? Overall, I am left with the impression that the author does not have sufficient familiarity

with the ice rheology literature.

p. 10, 26-27. "In summary, ice flow around a bedrock obstacle in temperate ice is constrained either by stoss-side pressure melting or by enhanced creep." Again this is misleading. Any obstacle is accommodated by both mechanisms, although one can dominate the other depending on bump size.

p. 10, 29-32 & p. 11, 1-4. None of these bullets, or the following assertion, has been demonstrated in this paper.

p. 10. 5-9. The conclusion here that temperature ice does not obey a power-law rheology has not been demonstrated in this paper.

p. 11, 19. The idea that no sliding occurs at subfreezing temps is not strictly correct. See Shreve 1984 J Glaciol.; Cuffey et al 1999, GRL.

Section 8.2. This is an interesting story but no aspect of it has been demonstrated in this paper.

p. 12, 8-9. "The corollary of the processes described herein is that if a thick temperate layer is present, basal motion over a hard bed with bedrock humps provides less drag than previously thought." This is misleading. The fact that traditional sliding theories, including Weertman's, under-predict rates of glacier sliding and over-predict drag has been discussed for many decades, and is well described in the leading reference book by Cuffey and Paterson (2010), for example. Under-emphasized in this paper is the role of cavity formation in reducing basal drag (e.g., Schoof, 2005).

p. 12, 11. "weaker bulk rheology"? I can see how ice can be "weak", but I don't understand how the relationship between stress and strain rate (rheology) can be "weak".

Section 8.3. The problem of fast flow on a hard bed is indeed important and may certainly involve soft ice and the mechanical and thermal effects of water flow under glaciers. The problem is that this paper does not provide new analyses (or data) that convincingly bear on the issue, so these comments on ice streaming come off as speculative and poorly motivated.

Conclusions. Not convincingly supported.

---

## Author Comment (AC5) · 26 May 2016

**Reply to interactive comment by  Anonymous Referee #3**

**on "Basal sliding of temperate basal ice on a rough, hard bed: pressure melting, creep mechanisms and implications for ice streaming" by M. Krabbendam**

Reviewer comments in black, reply in red.

Dear anonymous reviewer.

Thank you for the review.

Please note: below my detailed reply is a:

- Revised introduction (draft)

- Revised section on the "Creep Component in Temperate ice" (draft)

I am sympathetic to the motivation for this paper: Weertman sliding is indeed flawed in important ways, and heat transferred by subglacial water flow and produced by debrisbed friction may affect sliding physics. More broadly, the problem of how glaciers slide rapidly over hard, rough beds is an important one.

Thank you.  I wish to stress that this paper is conceptual in character, and arguably poses more questions and hypotheses than it answers, but these are questions that need be answered and hypotheses to be tested if the science is to move forward.  This is now emphasised in the revised introduction.

This paper, however, has serious deficiencies:

1) Most importantly, it contains little new analysis or data that help shed light on sliding physics. For example, the calculation of p. 4 yields conclusions that could have been reached without the calculation (see below, comments on p. 4, 1-21) and is used inappropriately to assert that the Weertman model is "illogical" (p. 4, 23-26).

MK: I beg to differ.  It contains an important element of consideration concerning frictional heat production, which is widely ignored; it reviews what little is known about temperate ice, and questions, with good reason, the use of Glen's flow law for such ice; it proposes a different way of heat transfer for pressure melting, and hence suggests that ice sheet modelling need to take into account 3 rather than 2 thermo-mechanical modes of sliding.  As the reviewer does not state that this has been done before, these analyses (conceptual and qualitative as they are) are new.

and is used inappropriately to assert that the Weertman model is "illogical" (p. 4, 23-26).

*MK: Fair point:  I will rephrase this as: "This result contradicts most observed geomorphology (Stokes and Clark, 1999; Bradwell et al., 2008) and supports the notion that pressure melting is not important for large obstacles".*

Too much of the paper consists of inferences not supported by data or relevant formal analysis.

2) Misconceptions/errors indicate a muddled understanding of relevant physics related to sliding (e.g., p. 7, 5-7; p. 7, 8-11; p. 8, 26; p. 10, 26-27; p. 11, 19) and ice rheology

(e.g., p. 9, 8-15).
MK: some poor phrasing on my part, but I reject the term 'muddled understanding of relevant physics':

- p. 7, 5-7; p. 7, 8-11.  No, this is entirely logical .  The question is whether water flow is sufficient – and that needs observations to confirm – see below.
- p. 8, 26.  Merely poor phrasing on my part.  Will change to: "albeit enhanced by stress concentration near the obstacle".
- p. 10, 26-27;  Merely poor phrasing on my part, see below.  :
- p. 11, 19: Can't think of what is muddled here, unless the reviewer refers to the negligible (negligible!!!) contribution of cold-based sliding – see further below. The remainder is mainstream, accepted glaciology.
- p. 9, 8-15 I will change this section, as per comments to M Montagnat (reviewer)

The attempt to assess the extent to which temperate ice obeys
a power-law flow rule by considering the dependence of strain rate on temperature,
rather than on stress, is a particularly major error.

MK: Not it is not.  The reviewer needs to realise that the Arrhenius relation is part and parcel of a power law in most if not all materials, and certainly ice (Glen 1955). Nevertheless, I've probably phrased this too strongly too early on, but the reviewer also misses the point that later the stress dependency is dealt with, with good evidence in some circumstances for n~1.  I've now restated that the Arrhenius temperature relationship does not work anymore, but also provided more examples of experiments/observations that show that n < 3, i.e. a departure from the 'normally assumed' n=3. The main point is that temperate ice behaves fundamentally different from cold ice.  With the sudden strain rate increase at -0.2°C AND evidence for n~1 behaviour, I feel this is justified.  It is unreasonable to change the Activation energy by some 'fudge factor' if there's a fundamental and significant change in rheological behaviour near the melting temperature. Thus, temperate ice does not follow a standard power-law creep behaviour, and it would be irresponsible to claim that it does if there's evidence to the contrary.  See new section on 'Creep component' at the end of this reply.

3) References are used inappropriately to support conclusions (p. 9, 8-15; p. 9, 8-23).
*MK: p. 9, 8-15  I have strengthened this, and provided more references that show that the stress dependency in many cases is close to 1.  I do not see how this is inappropriate.* See new section on 'Creep component' at the end of this reply.
p. 9, 8-23: I do not understand what the reviewer refers to, but if so probably covered in the new section on 'Creep component' at the end of this reply..

4) Inadequate justification is provided for some of the paper's assumptions (p. 5, 2; p.7, 5-7; p. 8, 1-4; p. 8, 5)
*MK: p. 5, 2 . "The normal vertical stress σnv can be taken as the effective pressure"  Explained in detail below*

*MK: p.7,5-7;  "If sufficient water is flowing through the system, heat advection by flowing water will be much more efficient than heat conduction through rock or ice":  As such this is true, but the question is whether water flow is sufficient – fair point, will explain better.  I will rephrase – see more below.*

MK p. 8, 1-4; *" Increase of basal water pressure Pw, resulting in a drop in effective pressure Pe , lowering the friction on flat surfaces. Frictional heating and drag on the flats will drop, as long as Pw remains high. Because there is less drag on the flat surfaces, the normal stress σn stoss onto the stoss side of obstacles, however, increases (also temporarily), enhancing stoss-side melting and basal melting".* These are simple physic/mechanical principles, frankly.  But I will change the last part into: "… enhancing both stoss-side melting as well as enhanced creep".

MK: p. 8, 5. Explained in detail below

5) The introduction would benefit from substantial revision. The attempt to motivate the subject of this paper (i.e., sliding mechanics) with bullets 1-5 is not successful (see my comments below on p. 2-3).

MK: Fair point, in a sense the paper explores conceptually  the potential effects of a temperate layer of significant thickness below cold ice in fast flowing ice, rather than just focussing on 'Weertman Sliding' : then the bullet points are appropriate as they show both the existence of that layer and the relevance to real world glaciological and geomorphological problems.

A draft of e new a revised Introduction is at the end of this reply.

**Specific comments keyed to page and line numbers:**

p. 1, 14-15. "Thermal equilibrium" is vague. There is "thermal equilibrium" in Weertman's model. The intended meaning needs to be clarified.

MK: Oh dear.  Weertman's model relies on a thermal gradient between the stoss and the lee-side, so there is no thermal equilibrium.  Muddled thermodynamics, I'm afraid. Thermal equilibrium has a very clear meaning, i.e. having a uniform temperature distribution.  (I do of course state later on there is thermal gradient near the stoss-side only, but this is the abstract, so 'near-thermal equilibrium' is a good enough). No reason to change.

**MAIN TEXT**

p. 1, 18. Power Law Creep should not be capitalized.   MK.  Fair point.  It may need a hyphen: 'power-law creep'.

p. 2, 13-15. "The essence of Weertman's sliding model is that basal ice movement past an obstacle is controlled either by stoss-side pressure melting around the obstacle or by ductile flow enhanced by stress concentrations near the obstacle, whichever is the fastest." This wording is a bit misleading. It suggests that either pressure-melting or ductile flow occurs, when regardless of bump size, there will always be components of both, albeit with one more important than the other (except for the transition bump size for which they equally important).

MK: To be  replaced by: "The essence of Weertman's sliding model is that basal ice movement past an obstacle occurs by stoss-side pressure melting around the obstacle and by ductile flow enhanced by stress concentrations near the obstacle, with ductile flow being more important with larger obstacle size."

p. 2. 25-29. It is not clear why fast ductile flow and soft ice somehow contradict Glen-type power law flow (i.e., the nonlinear dependence of strain rate on stress).

MK fair point, although in the end it is true (because temperate ice does not behave according to classic Glen's flow law, with both a breakdown of the Arrhenius relationship AND a departure from n=3). , but this is explained – see also 2 points below. Arguably, the paper is more about the effects of a temperate ice layer on sliding and deformation in fast flowing ice on hard, rough beds (e.g.hard-bedded ice streams), which includes the effects on the sliding mechanism. I will change the Introduction accordingly – see Revised Introduction at the end of the reply.

p. 2-3. 30-4. Although these are valid observations, the author needs to be more explicit about how they contradict pressure melting or a power-law ice rheology.
MK see above point, I will rephrase the introduction. Revised Introduction at the end of the reply

p. 3 12-18. These comments on basal thermal regime and basal hydrology in Greenland come as a surprise because there is no allusion to Greenland earlier in the introduction. Such an allusion is necessary because temperate ice of thicknesses much larger than bump size and water access to the glacier bed are, of course, normal for temperature glaciers. If sliding OF THE GREENLAND ICE SHEET, is how the author wants to motivate this paper, then that should be made clearer at the beginning of the introduction.
MK see above point, I will rephrase the introduction. Revised Introduction at the end of the reply

p. 3. Omit "worked" before "example" here and elsewhere.
MK: OK.

p. 4. 13. "was seen" indicates that this was actually observed. Express this differently here and where used elsewhere.          MK: changed to 'was regarded'

p. 4 "is then", earlier "was seen". Here and elsewhere the author tends to change tense in midstream.
MK: bearing in mind my limited knowledge of English grammar, I feel that in this case it is justified:  X was regarded to be a function of Y; this means that A, B.  But I'll check the tenses.

p. 4. 1-21. The reason for this calculation is unclear. The points made after it– that cavity formation can impede heat transport and that regelation speed decreases linearly with bump length–could be made without presenting the calculation. Even if there were better motivation for presenting this calculation, the source of numerical values, such as those for ice thickness and bed shear stress, is unclear (not in the text or appendix but presumably from somewhere in Greenland) and the choice of bump size and spacing is seemingly arbitrary.
MK:  I see your point.  The main point is to show that the thermal gradient set up through the obstacle is very small, so that other heat transport mechanisms are likely to be more efficient and hence more important.  But I will explain this better.

p. 4, 23-26. The author concludes at the end of the calculation: "This implies that ice flowing around an obstacle that is, say, four times longer than another obstacle (Fig. 1b), would be four times slower, even though this obstacle is more streamlined (having

a longer aspect ratio). This result is illogical, contradicts most observed geomorphology (Stokes and Clark, 1999; Bradwell et al., 2008), and is a major weakness of the Weertman model." This decrease in speed with elongation is a major weakness of the Weertman model ONLY if one neglects viscous flow in the Weertman model, as the author does here. And how can the author consider only pressure melting and refreezing–most relevant to bumps less than 0.5 m in wavelength–and assume that the calculation has relevance to the much larger landforms considered by geographers like Stokes and Clark? The Weertman model is indeed flawed, but this calculation adds nothing new to the subject, and the verbiage toward the end of the paragraph is misleading.

MK  Fair point.  Will rephrase as follows: "This result contradicts most observed geomorphology (Stokes and Clark, 1999; Bradwell et al., 2008) and supports the notion that pressure melting is not dominant  for large obstacles".

p. 5, 2. Explain why this approach is valid. In Hallet's abrasion model (1979; 1981) for example, debris-bed friction is independent of normal stress (and effective normal stress) and instead depends on the rate of ice convergence with the bed, so the assumption made here requires justification. Even for a flat bed, ice will converge with it due to basal melting, and that process exerts a downward drag on clasts, increasing friction between them and the bed. I think equation 5 can, in fact, be justified, but the necessary justification is not provided here.

MK.  What I've taken are the bulk friction coefficients from experiments (lab and subglacial) and which represent friction coefficients averaged over an area (eg. Budd et al, 1979; Zoet et al. 2013, Cohen et al. 2005) and then applied standard Coulomb friction, rather than the theoretical approach of Hallet, which depends on the contact friction coefficient at the (sparse) debris/bed contact points. Hallet's model is somewhat different, as it looks at abrasion by clast-bed wear rather than friction (pure ice over bedrock gives friction but negligible abrasion of the bedrock).  Therefore, Hallet focussed on the clast-bed contact forces, and these are indeed in his model dependent on ice convergence to the bed rather than the normal stress.  Note that the friction coefficient used in the Hallet model is the rock-rock friction coefficient (in the order of mu = 0.5-0.6) at the debris-bed contact spot (which is likely to be quite small), rather than the bulk ice / bed friction coefficient that results from the experiments

Note further that Budd et al. (1979) experiments were for clean ice, so a standard Coulomb friction law is applicable.  Only Budd et al. (1979)  varied the normal stress, so the other experimental work cannot be used to validate/invalidate the notion that normal stress has no effect.  Budd et al. (1979) reported a c. 6 x increase in wear at constant sliding velocity associated with a doubling of the normal stress, suggesting that normal stress *does* play a role.   Possibly, the disparity between Hallet's theory and the empirical observations arises because if debris is present, some of it will be in contact with the bed regardless of the convergence rate (if only due to negative buoyancy forces) combined with the effect that in fast moving ice, the sliding rate is orders of magnitude greater than the convergence rate.  A further possibility is that the contact force of a clast is a combination of the Hallet model (independent of normal stress) and the Boulton (1974) model (proportional to normal stress).

I will add a couple of sentences summarising the above, to better justify the standard Coulomb friction model.

p. 6. 11. A temperate ice layer, in fact, has a thermal gradient–one that reflects the decrease in melting temperature with pressure, as pictured in Figure 1 of this paper. Rewrite.

MK: rewritten to: "a temperate ice layer has no bulk thermal gradient (it has arguably a very small negative gradient, but this is ignored here), so no heat can be conducted through it; it forms a near-ideal thermal barrier (e.g. Aschwanden and Blatter, 2005)". Key here is that the normal increase in temperature cannot be sustained by conduction alone, the slope of the gradient (compared to the geothermal gradient below and the effective gradient (sustained by advection) above) is miniscule and opposite. This should now be obvious. (For information: the temperature gradient is c. 0.0007 °C/m or 0.7 °C/km)

p. 6. This page-long digression ("Intermezzo") distracts from the theme of the paper and will leave readers wondering what this paper is supposed to be about.

MK: Fair point. I will move this forward, to explain more about how a thick temperate layer can grow and be maintained. This is a serious problem, but by dealing with this first, it is not an intermezzo anymore. I plan to reorganise the revised MS as follows (see also my reply to D Cohen's comments:

1 Introduction
2 Basal meltwater production by frictional sliding
3 Growing and maintaining a temperate ice layer
4 The creep component in temperate ice (this will be expanded to take care of M Montagnat's comments)
5 The pressure melting component
6 Stoss-side pressure melting in temperate ice
7 Effect of surface water input on temperate ice on a rough bed
8 Critical obstacle size
9 Discussion
10 Conclusions

p. 6 34. This is a melt rate rather than a flux, which begs the question why melt rate was not expressed in Equation 6 with these units, as it normally is, either by defining the heat of fusion volumetrically or including density.

MK: 1) No, it is not the melt rate, although the melt rate puts an upper bound on to the flux. However, it is possible that some/much of the meltwater produced by frictional heating at the base is NOT percolated through the temperate layer towards the CTB, but flows towards the terminus and (eventually) escapes through a subglacial drainage system. So there's a difference between meltwater production (melt rate) and water flux through the temperate layer

2) the unit point is valid, and I will rejig equation (6) in SI units.

p. 7. 5-7. **(A)** The assertion here that heat flow through advection by flowing water is "more efficient" than heat flow due to thermal gradients in rock may be correct but is not demonstrated here or later in this paragraph. **(B)** Also, if some of the frictional and

geothermal heat is advected by water, why doesn't equation 6 reflect that? It should have a heat sink term in it associated with advection by flowing water.

(A) Fair point, see reply to next point.
(B) Equation (6) describes the melting rate of over a very large area, whereas the advection here merely transports (potentially) heat from one spot to another (from bump to bump) within the larger overall system. Similar small thermal gradient will also for instance occur at the high-friction clast/bed contacts, which may locally and possibly short-lived have high temperatures above the melting point on the very small ($mm^2$) contact areas….

p. 7. 8-11. The physics here is muddled. The ice temperature will be pinned everywhere along the bump surface at values set by the distribution of ice pressure (unless the temperature of the water in the film that divides ice from rock is not in equilibrium with the ice temperature, which seems unlikely). Although the water flow will cause some extra melting, if lee-side cavities do not form, the thermal gradient will be set by the pressure deviation from hydrostatic on both the stoss and lee sides of the bump. For a reason I don't understand, the author is assuming pressure is only important on the upstream side (see Figure 1c also).

MK: Mmmm, the key here is a) we' re in a net-melting environment, with excess water and hence high Pw; b) cavities may occur; c) therefore no net-negative deviatoric stress can be set up on the lee-side and the pressure there will be controlled by Pw, rather than the deviatoric stress. In that case (deviatoric) stress (not pressure!) is indeed only important at the stoss-side. I will reword to:

"In our conceptual model the temperate layer is thicker than the height of the obstacle. Water is continually produced by frictional heating, there is a net-melting environment with an excess of water, and water pressure on the ice-bed contact will be high. Water likely flows in a film and/or small gaps between bumps and obstacles, and cavities filled with water are likely to form. If sufficient water is flowing through the system, heat advection by flowing water can be as efficient if not more so than heat conduction through rock or ice. Consequently, no significant thermal gradients can build up and the entire basal system (temperate ice, water, and top rock) is held at *Tm*. "

Also the word "cold patch", as used in the classic paper by Robin (1976) and by subsequent textbook writers (e.g. Hooke), describes ice below the pressure melting temperature (PMT). To use it in this context, where all ice is at the PMT, is thus confusing.

MK: Not at all: *"The only exception (to PMT) is the stoss-side of a bedrock obstacle, where the melting temperature is continually depressed as a result of the concentrated deviatoric stress acting onto it"* this is the same as the use of Robin (1976), and is perfectly clear and not confusing at all. The other reviewer suggested to think of a different term, as cold patches can be taken to mean frozen patches in a polythermal situation

p. 8, 1-4. Debris-bed friction can be affected by water flux to the bed only if water can gain entry to zones where there may be small cavities beneath debris particles. This requires moving water from the channel at the point of entry to the bed out through the thin film that divides debris-bearing ice from bedrock. This propagation of pressure will be diffusive and slow, so it is unclear how much a sudden increase in warm meltwater will really decrease effective pressure and thereby reduce frictional drag.

MK:  This is the increase in Pw that occurs locally near to the point to where supraglacial lake drainage events occur. It has been observed that this can lead to minor but measurable local uplift (das et al. 2008; Hoffman et al. 2011), suggesting that, despite what the reviewer asserts, Pw increases do occur and quite rapidly so, although they maybe localised and short-lived.

p. 8, 5. The thermal gradient towards stoss surfaces depends not only on the temperature of the incoming warm water but on the distance between it and the stoss surfaces of bumps. Because the distances between channels carrying water and stoss surfaces is poorly known and could be far larger than the distance across a bedrock bump, the importance of this effect for stoss-side melting is uncertain, contrary to the certitude of the statement made here. (B) Perhaps the author is assuming that all of the warm water in a Das-like event moves in the thin film that divides ice from rock. If so, that is a dubious assumption.

MK:  I do not assume this (all of the warm water in a Das-like event moves in the thin film), but the reviewer assumes that all water in a 'Das-like' event will flow through channels, which is even more unlikely: it is clear from the scale of these events, and the measured vertical uplift that the subglacial channel system cannot carry all the water away, and thus some will develop a film.   I agree that the effect and importance are poorly known, and will put a caveat in.

p. 8, 26. "Weertman (1957) assumed that the creep component of ice flowing around a hard obstacle worked with a rheology" according to 'Glen's Flow law', albeit enhanced by stress concentration on the stoss side." Again here, the author makes the error of assuming deviatoric stresses are concentrated only on the stoss sides of bumps in Weertman's theory. Rather, deviatoric stresses are symmetric across bumps in the theory, with lee-side deviatoric stresses equal to but opposite in sign of those on stoss surfaces.

MK: will rephrase as:   " ….  albeit enhanced by stress concentration around the obstacles"

p. 8, 27. "strain-rate" rather than "strain".

MK: It reads:  "strain rate".  I'm not sure if these needs a hyphen, it is not the case that two nouns modify a third.  No need to change.

p. 8, 28. Power Law should not be capitalized, here and elsewhere.  MK: OK.

p. 9, 3. "comparisons" "suggests" Correct subject-verb correspondence.

MK: good point, corrected.

p. 9, 9. In fig. 3 strain rate is plotted, not strain as reported here.

MK: It reads:  "strain rate", not clear what the reviewer refers to.

p. 9, 8-15. Here the author asserts that if the log of strain rate plots as a straight line against the reciprocal of temperature, then power law creep is indicated. He needs to be aware that power-law creep is defined on the basis of the relationship between stress and strain rate, rather than between stress and temperature. Note that Morgan (1991) (the source of the data reproduced here) never commented on whether his data

conformed to power law creep rules because all of his tests were done at a single stress (0.1 MPa).

MK:  I will reword this into: "Thus, at constant stress, ice above c. -0.2°C shows a sudden weakening of a factor 5 to 10, and the Arrhenius relation of equation (7) clearly does not describe this, potentially implying that Power Law Creep is not dominant close to the melting temperature (see also Barnes et al. 1971; Colbeck and Evans, 1973; Morgan, 1991). "

The reviewer needs to be aware that the Arrhenius relation is part & parcel of the power law (certainly for ice) , so that if the Arrhenius relation breaks down suddenly it is only correct to start questioning power law.  Clearly something is happening…

I then have provided more evidence from the literature that the stress dependency is NOT 3, but in many cases is close to n = 1.  Thus, power-law creep 'breaks down', but I've phrased this differently (although any metallurgist would agree with me..).

p. 9, 8-23. This point of the paragraph–that temperate ice obeys a near Newtonian flow rule–could conceivably be correct, given the relative lack of work on warm ice, but is not convincingly argued here. Other authors cited, such as Byers et al (2012) and Chandler et al (2008), did indeed suggest values of n near 1.0 but were careful to attribute those low values to low deviatoric stresses, for which there is some micromechanical justification for low n. For ice near glacier beds, however, and particularly near bumps, deviatoric stresses will tend to be high, and thus justification for low values of n is weak.

MK: I have rewritten this, and added more evidence from the literature.  The reviewer has a point with the low-stress / high stress, and this is now described.  However, there is ample evidence that the stress dependency in both experiments and in nature is NOT the 'standard' n=3 that is usually assumed.  Note that Chandler's work IS on a real glacier.  The main point is that glacial modellers should not assume  n = 3 in temperate ice, if there's no evidence.  See revised section on the "Creep Component in Temperate ice"

p. 9, 31. Why does the abbreviation, GBPS, not coincide to the first letters of "grain boundary pressure melting"? And why choose a new term here when there is ample discussion of this sort of deformation in the literature, much of which is not cited here? Overall, I am left with the impression that the author does not have sufficient familiarity with the ice rheology literature.

MK:  I have rewritten this, with more emphasis on dislocation creep & recrystallisation enhanced by the presence of melt, discussed the potential role of Grain Boundary Sliding, and used the more traditional term 'grain boundary melting' / 'internal pressure melting'. There is again much evidence for partial melting in deforming temperate ice, and I've strengthened this.  As it is unclear as to which mechanism is dominant (I admit I was pushing too hard for a single mechanism), I use the term 'melt-assisted creep' as a bucket term. The problem is that, to my knowledge, no-one has studied the stress-strain behaviour AND the microstructures, which can tell us about the deformation mechanisms.  See revised section on the "Creep Component in Temperate ice"

p. 10, 26-27. "In summary, ice flow around a bedrock obstacle in temperate ice is constrained either by stoss-side pressure melting or by enhanced creep." Again this is misleading. Any obstacle is accommodated by both mechanisms, although one can

dominate the other depending on bump size.

MK: not so much misleading, as poorly phrased. " Ice  flow around a bedrock obstacle in temperate ice is accommodated  by stoss-side pressure melting or by enhanced creep, with creep being more important for larger obstacles.  In temperate ice, the creep component is….[    ].  And the pressure melting is  […. ]"..

p. 10, 29-32 & p. 11, 1-4. None of these bullets, or the following assertion, has been demonstrated in this paper.

MK:  Note:   " it is proposed here that stoss side pressure melting is constrained by ….".  Given the preceding matter, these is a very reasonable propositions to make, that is potentially testable.  It is clear from the English that I oppose a hypothesis here, rather than demonstrate or assert.

p. 10. 5-9. The conclusion here that temperature ice does not obey a power-law rheology has not been demonstrated in this paper.

MK: It is now convincingly shown that temperate ice cannot be responsibly modelled according to a standard power-law rheology with the standard n=3.  What it should be modelled as maybe unclear, but part of the point of this paper is to point out where and why temperate ice is important, but that there are large gaps in the knowledge of that material.  It appears to me that the reviewer thinks that temperate ice behaves the same as cold ice, and can be modelled in the same way.  This is clearly not true.   Power law cannot be regarded as operating in temperate ice.

p. 11, 19. The idea that no sliding occurs at subfreezing temps is not strictly correct.
See Shreve 1984 J Glaciol.; Cuffey et al 1999, GRL.

MK: that maybe true, especially near cold/warm boundaries.  However, both these papers admit that compared to normal, let alone fast warm-based sliding velocities, the sliding is 'negligible', and therefore as an overall approximation, the scale of which is clear here, it is correct.

Section 8.2. This is an interesting story but no aspect of it has been demonstrated in this paper.

MK: The first part is mainstream glaciology, the parts of 3) and 4) follow logically from the paper.  A dismissive comment, that is not very useful for an author to act upon. However, I will rephrase the introductory line as: " In such a model, three thermomechanical basal regimes may thus occur,  with a potential fourth operating seasonally".

p. 12, 8-9. "The corollary of the processes described herein is that if a thick temperate layer is present, basal motion over a hard bed with bedrock humps provides less drag than previously thought." This is misleading. The fact that traditional sliding theories, including Weertman's, under-predict rates of glacier sliding and over-predict drag has been discussed for many decades, and is well described in the leading reference book by Cuffey and Paterson (2010), for example. Under-emphasized in this paper is the role of cavity formation in reducing basal drag (e.g., Schoof, 2005).

MK:  I assume that the reviewer refers to the phrase 'previously thought'; if so, fair point.  Rephrased as: "The corollary of the processes described herein is that basal motion of temperate ice over a hard bed with bedrock humps provides considerably less drag than if the ice is modelled with cold ice properties".

p. 12, 11. "weaker bulk rheology"? I can see how ice can be "weak", but I don't understand how the relationship between stress and strain rate (rheology) can be "weak".

MK: Rheology is the general study of plastic/viscous deformation, so has a rather wider meaning than merely the stress / strain rate relation – it includes such important things as temperature for instance. In some disciplines, this is then (possibly erroneously) used in term of 'rheological behaviour' and hence the 'rheology of material X'. I will rephrase this as: "Instead they suggest a different, weaker bulk rheological behaviour"

Section 8.3. The problem of fast flow on a hard bed is indeed important and may certainly involve soft ice and the mechanical and thermal effects of water flow under glaciers.

MK: exactly, that's part of the point of the paper

The problem is that this paper does not provide new analyses (or data) that convincingly bear on the issue, so these comments on ice streaming come off as speculative and poorly motivated.

MK: I stated right in the introduction that this paper is conceptual in character. It does provide new concepts that do bear on the issue, because, to my knowledge, no-one has invoked a thick temperate layer as an explanation for the NEGIS, nor an explanation for how palaeo-ice streams with poor topographic steering operated during the Pleistocene. As I understand, the fast flow of the NEGIS is explained either by very high geothermal heat flow that is geologically unreasonable, and for which the presented geophysical evidence (gravity, aeromag) is unconvincing, or by the presence of soft sediment, which is at odds with the rough nature of the bed, as well as with the observations that hard gneiss in deglaciated areas is normally free of till where significant ice sliding has occurred during Pleistocene (see also my Reply to O Eisen). Thus, there is currently no satisfactory explanation for the observed fast ice flow in the NEGIS. Yes, the existence of a thick temperate layer is speculative, but so are all previous explanations put forward. Science cannot move forward if hypotheses cannot be proposed: as long as they are testable. This hypothesis is eminently testable (by drilling or doing radar echo sounding with an appropriate frequency): whether it is correct, time will tell. I feel it is justified to suggest the possibility of the presence of a thick layer of temperate ice at the base of the NEGIS. Speculative: yes, but in a responsible way of posing a testable hypothesis; poorly motivated: no, quite the opposite.

Conclusions. Not convincingly supported.

MK: How interesting that the two other reviewers did not pick this up. I cannot escape the impression that the anonymous reviewer is a bit dismissive of alternative approaches and just 'does not like it. It would be a bit more helpful if the reviewer had indicated in a bit more detail which of the conclusions are not convincingly supported and why not. Yes, this paper arguably poses more questions than it answers, but I believe these are pertinent questions that need to be put forward, rather than buried under a blanket..

Draft rewrite of the Introduction and the Creep are below:

**1 Introduction  (new version)**

The manner in which ice deforms within an ice sheet and moves or slides over its base are critical to accurately model the dynamic past, present and future behaviour of such ice bodies (e.g., Marshall, 2005). The internal deformation of cold ice is fairly well understood, and predictions made on the basis of physical laws (e.g., Glen's flow law) are broadly confirmed by observations (e.g., Dahl-Jensen and Gundestrup, 1987; Paterson, 1994; Ryser et al., 2014, but see Paterson (1991) for problems with dusty ice, and Hooke (1981) for a general critique).   This is not the case for basal sliding, for which many parameters are poorly constrained. Instead, many models of modern ice sheets use an empirical drag factor or slip coefficient, derived from observed ice velocity and estimated shear stresses (e.g., MacAyeal et al., 1995; Gudmundsson and Raymond, 2008; Ryser et al., 2014). Using an empirical slip coefficient is reasonable to describe and understand present-day near-instantaneous ice sheet behaviour, but cannot reliably predict or reconstruct ice velocities if parameters such as ice thickness, driving forces and meltwater production change significantly.

This problem is particularly acute for ice streams with poor topographic steering. For such ice streams it is commonly assumed that the necessary low drag can be explained by the presence of soft sediment or deformable till (e.g., Alley et al., 1987; Hindmarsh, 1997; Winsborow et al., 2010), which has indeed been shown to occur below some ice streams in West Antarctica (e.g. Alley et al. 1986; King et al. 2009) and also in the geomorphological record (e.g. Margold et al. 2010).  However, there is increasing geomorphological evidence for palaeo-ice streaming on rough, hard bedrock-dominated beds without clear topographic steering. Hard, rough beds are widespread on the beds of the former Pleistocene ice sheets and also likely beneath the present-day Greenland and Antarctic ice sheets (e.g., Kleman et al., 2008; Eyles, 2012; Rippin, 2014; Krabbendam and Bradwell, 2014; Krabbendam et al., 2016).  Evidence for paleo-ice streaming has been reported from the former Pleistocene Laurentide and British ice sheets and deglaciated parts of West Greenland (Smith, 1948; Stokes and Clark, 2003; Roberts and Long, 2005; Bradwell et al., 2008; Eyles, 2012; Bradwell, 2013; Eyles and Putkinen, 2014; Krabbendam et al., 2016). In these areas, the deforming-bed models cannot apply because little or no soft-sediment is present. These palaeo-ice stream zones are surrounded by areas also subjected to less intense warm-based ice erosion suggesting intermediate ice velocities (e.g., Bradwell, 2013), consistent with ice velocity analysis and borehole observations from the Greenland Ice Sheet that show significant warm-based sliding (10-100 m yr-1) outside ice-streams (Lüthi et al., 2002; Ryser et al., 2014, Joughin et al., 2010). Thus, fast ice flow appears to be possible on hard, rough beds and cannot be explained by a simple cold/warm thermal boundary (cf. Payne and Dongelmans, 1997).  In Greenland, the massive Northeast Greenland Ice Stream remains difficult to explain, as current explanations invoke geologically unreasonably high geothermal heat flows (e.g., Fahnenstock et al., 2001) and a deformable bed with an unknown till source (Christianson et al. 2014).

A solution may be presented by the occurrence of a basal layer of temperate ice (ice at the melting temperature), below cold ice that makes up the remainder of the ice sheet. Drilling in Greenland Ice Sheet adjacent to the Jakobshavn Isbrae has documented a c. 30 m thick basal layer of temperate ice below cold ice (Lüthi et al., 2002), and has been modelled to occur beneath other parts of the Greenland Ice Sheet (e.g., Dahl-Jensen, 1989; Calov and Hutter, 1996; Greve, 1997). Two pertinent questions follow from these observations:

1) How does such a temperate layer develop and how is it maintained, given that it is overlain by cold ice? In-situ measurements at a glacier base and experiments have shown that warm-based basal sliding occurs under significant friction, caused by basal-debris / bedrock contacts (Iverson et al., 2003; Cohen et al., 2005; Zoet et al., 2013), generating significant frictional heat at the base, which is important for the development of a temperate ice layer.

2) How does basal sliding work in temperate ice, and how does this differ from classic sliding models? The essence of the classic Weertman (1957) sliding model is that basal ice movement past an obstacle occurs by stoss-side pressure melting around the obstacle and by ductile flow according to Glen's flow law but enhanced by stress concentrations near the obstacle, with ductile flow being more important with larger obstacle size. Sparse experimental evidence suggests that temperate ice is considerably weaker than cold ice, and that creep may not be modelled reliably according to the standard Glen's flow law (e.g. Colbeck and Evans, 1973; Duval, 1977; Morgan, 1991). Secondly, in a temperate ice layer, the thermal gradients required for the pressure melting (e.g. heat flow *through* the obstacle) to proceed may have different controls than in the classic model.

This paper deals with four issues:

- The problem of how a temperate layer can develop below cold ice, including the role of frictional heating;
- How the basic assumptions of classic Weertman sliding (enhanced ductile flow controlled by Glen's flow law and stoss-side pressure melting controlled by heat flow through an obstacle) may not be applied to temperate ice, and alternative controlling mechanisms are proposed;
- The potential thermo-mechanical role of temperate ice below cold ice in an ice sheet;

[revised manuscript text omitted]
, 1997, 2001), as well as the development of a crystallographic fabric (Tison and Hubbard, 2000), suggests grain boundary sliding is probably not significant in temperate ice. Diffusion creep, grain boundary sliding and grain boundary melting all work on grain boundaries and are grain-size sensitive: they are favoured by a small grain size and the presence of a liquid; these mechanisms normally result in n < 3, and thus could explain the n ~ 1 behaviour seen in some

experiments and natural glaciers. Considering the near-unique pressure-melting behaviour of $H_2O$, grain boundary melting is worthy of further study. However, all grain-size sensitive mechanisms are at odds with the large grain sizes observed and can, on their own, not explain well-developed fabrics. Conversely, well-developed fabrics potentially attests to dislocation creep, but this at odds with the n ~ 1 behaviour commonly observed. Altogether there is no clear evidence of a single dominant deformation mechanism, and all deformation mechanisms mentioned above may contribute. The change in the stress-dependency as observed in the temperate Worthington Glacier (Marshall et al. 2005) as well as in some experiments (De La Chapelle et al. 1999) suggests that the dominant deformation mechanism in temperate ice depends on the magnitude of stress. For the moment the rather non-generic term '*melt-assisted creep*' is used herein, while stressing that regardless of the actual mechanism, all experiments show that temperate ice with high water content is significantly weaker than cold ice.

A strong crystallographic fabric and concentrations of dust or silt particles are known to significantly weaken cold ice in simple shear (e.g., Lile, 1978; Paterson, 1984, 1991; Dahl-Jensen and Gundestrup, 1987; Azuma, 1994), but whether this leads to further weakening of temperate ice is not known. There is still much unknown about creep in temperate ice; all that can be said is that temperate ice is significantly weaker than, and behaves very differently from, cold ice.

---

## Author Response (AR1)

**Reply to reviewers comments**

**Summary of changes.**

1) I have changed the organisation of the paper:
   a. I now first describe the heat production by frictional heating, and the problem of the growth and maintenance of a temperate layer (new Section 2 and 3). This takes care of the awkward 'intermezzo', highlighted by the Anonymous Reviewer.
   b. Then I deal first with the creep component, which is likely to be more important than pressure melting, as commented upon by D Cohen.
2) The section of creep (new section 4) has been completely rewritten, to take care of the comments of M Montagnat, and some comments of the Anonymous Reviewer.
3) I've added a short section on the critical obstacle size, that now stresses the importance of creep over that of pressure melting (new Section 8 + new Figure 6), which is indeed important as indicated by D Cohen.
4) I've also rewritten the Introduction, which now focusses now more on the general issue of temperate ice (e.g. comments by the Anonymous reviewer), and less of a 'criticism' of Weertman sliding (e.g. comments by D Cohen), which was not meant to be the case.

Note 1; My replies on the comments below are in red

Note 2: in the tracked changes below, the changes in the Reference list have NOT been tracked.

==========================================================================================

**Detailed reply to comments by D Cohen.**

It was not my intention at all to deconstruct or criticise Weertman's original model ; all I wanted to do is to point out it is not applicable to temperate ice with the original assumptions, which, unfortunately, are at times still perpetuated in probably inappropriate conditions. I have reworded and changed the text to be more respectful to Weertman's original model, which of course was pioneering, and has become classic.

I agree that, for larger-scale obstacles, creep is more important than pressure melting against the obstacles. I've changed the order of the manuscript to reflect this, and added a section on Critical Obstacle size to discuss this, and how this differs from the original Weertman model.

Page 1 last line: Probably "near steady-state situations" should be replaced withsomething like "near-instantaneous situations". **MK:** done.

Page 2 Line 10. I think the author meant to refer to Cohen et al 2005 and not Cohen etal 2006. **MK:** Reference to Cohen et al. 2005: corrected.

Page 2 Line 24. I would insert the word "either" before "pose problems" **MK:** Corrected

Page 3 Line 20. I think the mechanisms of sliding are clear: ice at the melting temperature contains water and water between ice and bedrock forms a thin lubricating layer with near zero shear resistance that allows ice in contact with the bedrock to have non-zero velocities. The question is how to quantify sliding and what glaciological parameters control it. **MK: Reworded. The actual sliding (over a flat area) is fairly clear, the basal motion over a rough surface is not.**

Page 4 Line 20. Equation 4. There is an typo/error. It should be: Vpm = Q_obs/(H_ice*rho_ice). **MK: Oops, indeed, corrected.**

Page 4 Line 24. Strictly speaking in regelation ice does not flow around the obstacle. That's the viscous part of motion. In

10 regelation ice melts on one side, the water flows to the other side and refreezes there. May be change wording. **MK: Reworded.**

"Page 5 Line 2. The vertical stress could even be higher than the effective pressure since, due to melting, there is a component of ice flow towards the bed that creates a vertical downward force on the debris. This force could be significant

15 and further increase basal friction".

 **MK: T**hat is strictly true, but that concerns the very localised contact stress of the debris onto the bed, according to the 'Hallet model;'. On this contact the friction coefficient would also be much higher, probably in the order of μ = 0.5 or so. Since I use the friction coefficient averaged over a large area, I have also used the vertical stress over a large area, rather than looking at the very localised stresses. See longer explanation in the reply to Anonymous Reviewer.

"Page 6 Line 11 (ii). Strictly speaking this is not true. There will be differences in temperature in temperate ice due to differences in stresses (if only with depth). These temperature differences will cause thermal gradients and heat fluxes (arguably small). These gradients will only serve to melt ice or freeze water. See Lliboutry 1993.".

**MK:** This is strictly true, but here I'm referring here to a bulk thermal gradient, capable of transporting significant heat from

25 the base to the CTB. This bulk thermal gradient is zero (or rather it is slightly negative, due to the increase in cryostatic pressure). I've reworded this, to make it clear.

Page 7 Line 10. The use of the words 'cold patch' is confusing. The ice is at the
melting temperature so it's not cold. I think the term cold patch should be restricted

30 to cold ice not ice at the melting temperature that is colder because under a higher
pressure. See also Figure 1. **MK:** Cold patch: good point: changed to 'cool/warm spot', also on Figure.
=======================================================================================

**Detailed reply to comments by M Montagnat.**

M Montagnat makes some very astute points, and as a result I've rewritten the section on creep. I have discussed more the role of different deformation mechanisms, including enhanced dislocation creep, dynamic recrystallisation and grain boundary sliding. I've stressed now that grain boundary melting is not a 'new' mechanism, but has in fact been observed before. The behaviour of temperate ice is NOT very similar to other geological materials and metals at high temperatures. No other such material (with the apparent exception of plutonium!) shows the near-unique behaviour of H2O, namely that the liquid phase is denser than the solid phase. This fundamentally different phase-transition behaviour is likely to lead to a fundamentally different deformation behaviour. In other words, the whole concept of pressure melting, namely that higher pressure leads to a lowering of the melting temperature only works for H2O. Both the dramatic and sudden increase in strain rate close to the melting temperature, and the (albeit limited) evidence for near-Newtonian (n ~ 1) behaviour in temperate ice do suggest that a) a power-law does not describe the behaviour (both in terms of the breakdown of the Arrhenius relation, as well as evidence for n ~ 1 behaviour – this is now described better; b) there is a switch in rate-controlling deformation mechanism(s), rather than 'merely' a variation of dislocation creep.

- p 9, 2d paragraph: I don't think that results on temperate ice show that power-law creep does not adequately describe ice creep above -0.2_C. With increase in temperature, the activation energy can change, and therefore, we do not expect a linear relation between strain rate and temperature. To get rid of a power-law creep relation, one needs to plot minimum strain-rate as a function of stress... and show that this is not linear. Which is not provided in Morgan 1991. Results by De La Chapelle et al tend to show that power-law creep remains even when a liquid phase exist at GB, which is mostly what happens when ice is temperate (Wilson et al. 1996)?

**MK:** I've now described better that a) the Arhenius relationship does not work **and** b) that the stress dependency changes. Note the La Chapelle experiments are not truly in temperate ice, but in an cold ice/brine mixture.

- p9, last paragraph: to my point of view, there are not enough information to be able to suggest a mechanism such as the one suggested here (GBPS), that has never been observed in ice, and that appears more than unlikely regarding knowledge about ice deformation behavior, with or without liquid layers... A discussion would nevertheless be required concerning the possibility of grain boundary sliding at such high temperatures. **MK:** grain boundary melting has been described before – this is now discussed better. The possibility of grain boundary sliding is now discussed, as well as the enhancement of dislocation creep.

- p10, 1st paragraph: As far as I know, pressure solution requires different phases to be present, in order for some to be dissolved under local pressure, and migrate is some fractures together with the fluid phase, and re-precipitate further away (see J-P. Gratier, D. K. Dysthe and F. Renard. The role of pressure solution creep in the ductility of theEarth's upper crust.

Advances in Geophysics, vol. 54, 2013). I do not see at all how this can occur in the temperate ice layer at the bottom of glaciers and ice sheets...

**MK:** The analogy lies in the fact that solid changes to a liquid and vice versa, and that this leads to strain. The analogy with pressure melting (and the difference) is now described better.

- p11, 1st paragraph: Once again, there are not enough proof or information to assess so directly the occurrence of some grain boundary pressure melting... Power-law creep can also be fast, if accommodation mechanisms are efficient (dynamic recrystallization, GBS, liquid intergranular phase)... so it can not be ruled out so easily.

**MK:** completely right, which is why I've rewritten the section and discussed all these deformation mechanisms. Grain
10  boundary sliding is, however, unlikely, due to very large grain sizes, as now discussed.

=============================================================================================

**Detailed reply to comments by Anonymous Reviewer**

This paper, however, has serious deficiencies:

15  1) Most importantly, it contains little new analysis or data that help shed light on sliding physics. For example, the calculation of p. 4 yields conclusions that could have been reached without the calculation (see below, comments on p. 4, 1-21) and is used inappropriately to assert that the Weertman model is "illogical" (p. 4, 23-26).

**MK:** I beg to differ.  It contains an important element of consideration concerning frictional heat production, which is widely ignored; it reviews what little is known about temperate ice, and questions, with good reason, the use of Glen's flow
20  law for such ice; it proposes a different way of heat transfer for pressure melting, and hence suggests that ice sheet modelling need to take into account 3 rather than 2 thermo-mechanical modes of sliding.  As the reviewer does not state that this has been done before, these analyses (conceptual and qualitative as they are) are new.

and is used inappropriately to assert that the Weertman model is "illogical" (p. 4, 23-26).

**MK:** Fair point, I've rephrased this.

2) Misconceptions/errors indicate a muddled understanding of relevant physics related to sliding (e.g., p. 7, 5-7; p. 7, 8-11; p. 8, 26; p. 10, 26-27; p. 11, 19) and ice rheology (e.g., p. 9, 8-15).

**MK:** some poor phrasing on my part, but I reject the term 'muddled understanding of relevant physics':

- p. 7, 5-7; p. 7, 8-11.  No, this is entirely logical .  The question is whether water flow is sufficient – and that needs
30  observations to confirm – see below.
- p. 8, 26.  Merely poor phrasing on my part.  Have change to: "albeit enhanced by stress concentration around the obstacle".
- p. 10, 26-27;  Merely poor phrasing on my part, see below.  :
- p. 11, 19: Can't think of what is muddled here, unless the reviewer refers to the negligible  (negligible!!!)
35  contribution of cold-based sliding – see further below. The remainder is mainstream, accepted glaciology.

- p. 9, 8-15 I have changed this section, as per comments to M Montagnat

The attempt to assess the extent to which temperate ice obeys a power-law flow rule by considering the dependence of strain rate on temperature, rather than on stress, is a particularly major error.

5  **MK:** Not it is not. The reviewer needs to realise that the Arrhenius relation is part and parcel of a power law in most if not all materials, and certainly ice (Glen 1955). Nevertheless, I've probably phrased this too strongly too early on, but the reviewer also misses the point that later the stress dependency is dealt with, with good evidence in some circumstances for n~1. I've now restated that the Arrhenius temperature relationship does not work anymore, but also provided more examples of experiments/observations that show that n < 3, i.e. a departure from the 'normally assumed' n=3. The main point is that

10  temperate ice behaves fundamentally different from cold ice. With the sudden strain rate increase at -0.2°C AND evidence for n~1 behaviour, I feel this is justified. It is unreasonable to change the Activation energy by some 'fudge factor' if there's a fundamental and significant change in rheological behaviour near the melting temperature. Thus, temperate ice does not follow a standard power-law creep behaviour, and it would be irresponsible to claim that it does if there's evidence to the contrary. This should now be clearer in the rewritten section on 'Creep component'.

3) References are used inappropriately to support conclusions (p. 9, 8-15; p. 9, 8-23).

*MK: p. 9, 8-15 I have strengthened this, and provided more references that show that the stress dependency in many cases is close to 1. I do not see how this is inappropriate.* See new section on 'Creep component' at the end of this reply.

p. 9, 8-23: I do not understand what the reviewer refers to, but if so probably covered in the new section on 'Creep

20  component' at the end of this reply..

4) Inadequate justification is provided for some of the paper's assumptions (p. 5, 2; p.7, 5-7; p. 8, 1-4; p. 8, 5)

*MK: p. 5, 2 . "The normal vertical stress σnv can be taken as the effective pressure" Explained in detail below*

25  *MK: p.7,5-7; "If sufficient water is flowing through the system, heat advection by flowing water will be much more efficient than heat conduction through rock or ice": As such this is true, but the question is whether water flow is sufficient – fair point. I have rephrased this*

MK p. 8, 1-4; " *Increase of basal water pressure Pw, resulting in a drop in effective pressure Pe , lowering the friction on*

30  *flat surfaces. Frictional heating and drag on the flats will drop, as long as Pw remains high. Because there is less drag on the flat surfaces, the normal stress σn stoss onto the stoss side of obstacles, however, increases (also temporarily), enhancing stoss-side melting and basal melting*". These are simple physic/mechanical principles, frankly. But have changed the last part into: "… enhancing both stoss-side melting as well as creep".

**MK:** p. 8, 5. Explained in detail below

5) The introduction would benefit from substantial revision. The attempt to motivate the subject of this paper (i.e., sliding mechanics) with bullets 1-5 is not successful (see my comments below on p. 2-3).

**MK:** Fair point, I've rewritten the Introduction.

**Specific comments keyed to page and line numbers:**

p. 1, 14-15. "Thermal equilibrium" is vague. There is "thermal equilibrium" in Weertman's model. The intended meaning needs to be clarified.

**MK:** Oh dear. Weertman's model relies on a thermal gradient between the stoss and the lee-side, so there is no thermal equilibrium. Muddled thermodynamics, I'm afraid. Thermal equilibrium has a very clear meaning, i.e. having a uniform temperature distribution = absence of a thermal gradient. (I do of course state later on there is thermal gradient near the stoss-side only, but this is the abstract, so 'near-thermal equilibrium' is a good enough). No reason to change.

**MAIN TEXT**

p. 1, 18. Power Law Creep should not be capitalized. **MK.** Fair point, rewritten as: 'power-law creep'.

p. 2, 13-15. "The essence of Weertman's sliding model is that basal ice movement past an obstacle is controlled either by stoss-side pressure melting around the obstacle or by ductile flow enhanced by stress concentrations near the obstacle, whichever is the fastest." This wording is a bit misleading. It suggests that either pressure-melting or ductile flow occurs, when regardless of bump size, there will always be components of both, albeit with one more important than the other (except for the transition bump size for which they equally important).

**MK:** Rewritten as: "The essence of the classic Weertman (1957) sliding model is that basal ice movement past an obstacle occurs by stoss-side pressure melting around the obstacle and by ductile flow enhanced by stress concentrations near the obstacle, with ductile flow being more important for larger obstacles."

p. 2. 25-29. It is not clear why fast ductile flow and soft ice somehow contradict Glen-type power law flow (i.e., the nonlinear dependence of strain rate on stress).

**MK:** this is now dealt with in detail in the new section on Creep.

p. 2-3. 30-4. Although these are valid observations, the author needs to be more explicit about how they contradict pressure melting or a power-law ice rheology.

**MK:** the introduction has been rewritten

p. 3 12-18. These comments on basal thermal regime and basal hydrology in Greenland come as a surprise because there is no allusion to Greenland earlier in the introduction. Such an allusion is necessary because temperate ice of thicknesses much larger than bump size and water access to the glacier bed are, of course, normal for temperature glaciers. If sliding OF THE GREENLAND ICE SHEET, is how the author wants to motivate this paper, then that should be made clearer at the beginning of the introduction.

**MK:** the introduction has been rewritten, and the motivation to deal with the NEGIS is now clear

p. 3. Omit "worked" before "example" here and elsewhere.

**MK:** OK.

p. 4. 13. "was seen" indicates that this was actually observed. Express this differently here and where used elsewhere.

**MK:** changed to 'was regarded'

p. 4 "is then", earlier "was seen". Here and elsewhere the author tends to change tense in midstream. **MK:** reworded

p. 4. 1-21. The reason for this calculation is unclear. The points made after it– that cavity formation can impede heat transport and that regelation speed decreases linearly with bump length–could be made without presenting the calculation. Even if there were better motivation for presenting this calculation, the source of numerical values, such as those for ice thickness and bed shear stress, is unclear (not in the text or appendix but presumably from somewhere in Greenland) and the choice of bump size and spacing is seemingly arbitrary.

**MK:** The main point is to show that the thermal gradient set up through the obstacle is very small, so that other heat transport mechanisms are likely to be more efficient and hence more important. I've taken the original Weertman parameters, for a fair comparison, as is clear from the text.

p. 4, 23-26. The author concludes at the end of the calculation: "This implies that ice flowing around an obstacle that is, say, four times longer than another obstacle (Fig. 1b), would be four times slower, even though this obstacle is more streamlined (having a longer aspect ratio). This result is illogical, contradicts most observed geomorphology (Stokes and Clark, 1999; Bradwell et al., 2008), and is a major weakness of the Weertman model." This decrease in speed with elongation is a major weakness of the Weertman model ONLY if one neglects viscous flow in the Weertman model, as the author does here. And how can the author consider only pressure melting and refreezing–most relevant to bumps less than 0.5 m in wavelength–and assume that the calculation has relevance to the much larger landforms considered by geographers like Stokes and Clark? The Weertman model is indeed flawed, but this calculation adds nothing new to the subject, and the verbiage toward the end of the paragraph is misleading.

**MK:**   Fair point.  Rephrased as: "This result contradicts most observed geomorphology (Stokes and Clark, 1999; Bradwell et al., 2008) and supports the notion that pressure melting is not dominant  for large obstacles".

p. 5, 2. Explain why this approach is valid. In Hallet's abrasion model (1979; 1981) for example, debris-bed friction is independent of normal stress (and effective normal stress) and instead depends on the rate of ice convergence with the bed, so the assumption made here requires justification. Even for a flat bed, ice will converge with it due to basal melting, and that process exerts a downward drag on clasts, increasing friction between them and the bed. I think equation 5 can, in fact, be justified, but the necessary justification is not provided here.

**MK:**   What I've taken are the bulk friction coefficients from experiments (lab and subglacial) and which represent friction coefficients averaged over an area (eg. Budd et al, 1979; Zoet et al. 2013, Cohen et al. 2005) and then applied standard Coulomb friction, rather than the theoretical approach of Hallet, which depends on the contact friction coefficient at the (sparse) debris/bed contact points.   Hallet's model is somewhat different, as it looks at abrasion by clast-bed wear rather than friction (pure ice over bedrock gives friction but negligible abrasion of the bedrock).  Therefore, Hallet focussed on the clast-bed contact forces, and these are indeed in his model dependent on ice convergence to the bed rather than the normal stress.  Note that the friction coefficient used in the Hallet model is the rock-rock friction coefficient (in the order of mu = 0.5-0.6) at the debris-bed contact spot (which is likely to be quite small), rather than the bulk ice / bed friction coefficient that results from the experiments

Note further that Budd et al. (1979) experiments were for clean ice, so a standard Coulomb friction law is applicable.   Only Budd et al. (1979)  varied the normal stress, so the other experimental work cannot be used to validate/invalidate the notion that normal stress has no effect.   Budd et al. (1979) reported a c. 6 x increase in wear at constant sliding velocity associated with a doubling of the normal stress, suggesting that normal stress *does* play a role.   Possibly, the disparity between Hallet's theory and the empirical observations arises because if debris is present, some of it will be in contact with the bed regardless of the convergence rate (if only due to negative buoyancy forces) combined with the effect that in fast moving ice, the sliding rate is orders of magnitude greater than the convergence rate.  A further possibility is that the contact force of a clast is a combination of the Hallet model (independent of normal stress) and the Boulton (1974) model (proportional to normal stress).

I have added a caveat,  to better justify the standard Coulomb friction model.

p. 6. 11. A temperate ice layer, in fact, has a thermal gradient–one that reflects the decrease in melting temperature with pressure, as pictured in Figure 1 of this paper. Rewrite.

**MK:**  rewritten as: "a temperate ice layer has no bulk  thermal gradient (it has arguably a very small negative gradient, but this is ignored here), so no heat can be conducted through it; it forms a near- ideal thermal barrier (e.g. Aschwanden and Blatter, 2005)".  Key here is that the normal increase in temperature cannot be sustained by conduction alone, the slope of

the gradient (compared to the geothermal gradient below and the effective gradient (sustained by advection) above) is miniscule and opposite. This should now be obvious. (For information: the temperature gradient is c. 0.0007 °C/m or 0.7 °C/km)

5  p. 6. This page-long digression ("Intermezzo") distracts from the theme of the paper and will leave readers wondering what this paper is supposed to be about.

**MK:** Good point, now covered by the re-organisation of the manuscript.

p. 6 34. This is a melt rate rather than a flux, which begs the question why melt rate was not expressed in Equation 6 with
10  these units, as it normally is, either by defining the heat of fusion volumetrically or including density.

**MK:**   1) No, it is not the melt rate *sensu stricto*, although the melt rate puts an upper bound on to the flux. It  is possible that some/much of the meltwater produced by frictional heating at the base is NOT percolated through the temperate layer towards the CTB, but flows towards the terminus and (eventually) escapes through a subglacial drainage system. So there's a difference between meltwater production (melt rate) and water flux through the temperate layer

15  2) the unit point is valid, and I have now shown the equation (now equation 2) in SI units.

p. 7. 5-7. **(A)** The assertion here that heat flow through advection by flowing water is "more efficient" than heat flow due to thermal gradients in rock may be correct but is not demonstrated here or later in this paragraph. **(B)** Also, if some of the frictional and geothermal heat is advected by water, why doesn't equation 6 reflect that? It should have a heat sink term in it
20  associated with advection by flowing water.

**MK:** Equation (6) (now equation 2) describes the melting rate of over a very large area, whereas the advection here merely transports (potentially) heat from one spot to another (from bump to bump) within the larger overall system. Similar small thermal gradient will also for instance occur at the high-friction clast/bed contacts, which may locally and possibly short-lived have high temperatures above the melting point on the very small (mm$^2$) contact areas….

p. 7. 8-11. The physics here is muddled. The ice temperature will be pinned everywhere along the bump surface at values set by the distribution of ice pressure (unless the temperature of the water in the film that divides ice from rock is not in equilibrium with the ice temperature, which seems unlikely). Although the water flow will cause some extra melting, if lee-side cavities do not form, the thermal gradient will be set by the pressure deviation from hydrostatic on both the stoss and lee
30  sides of the bump. For a reason I don't understand, the author is assuming pressure is only important on the upstream side (see Figure 1c also).

**MK:** The key here is a) we're in a net-melting environment, with excess water and hence high Pw; b) cavities may occur; c) therefore no net-negative deviatoric stress can be set up on the lee-side and the pressure there will be controlled by Pw,

rather than the deviatoric stress. In that case (deviatoric) stress (not pressure!) is indeed only important at the stoss-side. Reworded to:

"In the conceptual model here (Fig. 5c) , the temperate layer is thicker than the height of the obstacle. Water is continually produced by frictional heating, there is a net-melting environment with an excess of water, and water pressure on the ice-bed contact will be high. Depending on the basal melt rate and the amount of water flowing along the ice-rock interface, heat advection by flowing water may well be more efficient than heat conduction through rock or ice. In that case no significant thermal gradients can build up and the entire basal system (temperate ice, water, and top rock) is kept at thermal equilibrium at $T_m$ "

Also the word "cold patch", as used in the classic paper by Robin (1976) and by subsequent textbook writers (e.g. Hooke), describes ice below the pressure melting temperature (PMT). To use it in this context, where all ice is at the PMT, is thus confusing.

**MK:** Not at all: *"The only exception (to PMT) is the stoss-side of a bedrock obstacle, where the melting temperature is continually depressed as a result of the concentrated deviatoric stress acting onto it"* this is the same as the use of Robin (1976), and is perfectly clear and not confusing at all. The other reviewer suggested to think of a different term, as cold patches can be taken to mean (larger scale) frozen patches in a polythermal situation

p. 8, 1-4. Debris-bed friction can be affected by water flux to the bed only if water can gain entry to zones where there may be small cavities beneath debris particles. This requires moving water from the channel at the point of entry to the bed out through the thin film that divides debris-bearing ice from bedrock. This propagation of pressure will be diffusive and slow, so it is unclear how much a sudden increase in warm meltwater will really decrease effective pressure and thereby reduce frictional drag.

**MK:** This is the increase in Pw that occurs locally near to the point to where supraglacial lake drainage events occur. It has been observed that this can lead to minor but measurable local uplift (Das et al. 2008; Hoffman et al. 2011), suggesting that, despite what the reviewer asserts, Pw increases do occur and quite rapidly so, although they maybe localised and short-lived.

p. 8, 5. The thermal gradient towards stoss surfaces depends not only on the temperature of the incoming warm water but on the distance between it and the stoss surfaces of bumps. Because the distances between channels carrying water and stoss surfaces is poorly known and could be far larger than the distance across a bedrock bump, the importance of this effect for stoss-side melting is uncertain, contrary to the certitude of the statement made here. (B) Perhaps the author is assuming that all of the warm water in a Das-like event moves in the thin film that divides ice from rock. If so, that is a dubious assumption.

**MK:** I do not assume this (all of the warm water in a Das-like event moves in the thin film), but the reviewer assumes that all water in a 'Das-like' event will flow through channels, which is even more unlikely: it is clear from the scale of these events, and the measured vertical uplift that the subglacial channel system cannot carry all the water away, and thus some will develop a film. Nevertheless, I agree that the effect and importance are poorly known, and have rephrased to 'soften' the certitude.

p. 8, 26. "Weertman (1957) assumed that the creep component of ice flowing around a hard obstacle worked with a rheology" according to 'Glen's Flow law', albeit enhanced by stress concentration on the stoss side." Again here, the author makes the error of assuming deviatoric stresses are concentrated only on the stoss sides of bumps in Weertman's theory. Rather, deviatoric stresses are symmetric across bumps in the theory, with lee-side deviatoric stresses equal to but opposite in sign of those on stoss surfaces.

**MK:** rephrased as:  " …. albeit enhanced by stress concentration around the obstacles"

p. 8, 27. "strain-rate" rather than "strain". **MK:** changed

p. 8, 28. Power Law should not be capitalized, here and elsewhere. **MK:** OK changed.

p. 9, 3. "comparisons" "suggests" Correct subject-verb correspondence. **MK:**, corrected.

p. 9, 8-15. Here the author asserts that if the log of strain rate plots as a straight line against the reciprocal of temperature, then power law creep is indicated. He needs to be aware that power-law creep is defined on the basis of the relationship between stress and strain rate, rather than between stress and temperature. Note that Morgan (1991) (the source of the data reproduced here) never commented on whether his data conformed to power law creep rules because all of his tests were done at a single stress (0.1 MPa).

**MK:** Reworded as: "Thus, at constant stress, ice above c. -0.2°C shows a sudden weakening of a factor 5 to 10, evidently not described by the Arrhenius relation of equation (7), potentially implying that power-law creep is not dominant close to the melting temperature  (see also Barnes et al. 1971; Colbeck and Evans, 1973; Morgan, 1991). "

The reviewer needs to be aware that the Arrhenius relation is part & parcel of the power law (certainly for ice), so that if the Arrhenius relation breaks down suddenly it is only correct to start questioning power law. Clearly something is happening…

I then have provided more evidence from the literature that the stress dependency is NOT 3, but in many cases is close to $n = 1$. Thus, power-law creep 'breaks down', but I've phrased this differently (although any metallurgist would agree..).

p. 9, 8-23. This point of the paragraph–that temperate ice obeys a near Newtonian flow rule–could conceivably be correct, given the relative lack of work on warm ice, but is not convincingly argued here. Other authors cited, such as Byers et al

(2012) and Chandler et al (2008), did indeed suggest values of n near 1.0 but were careful to attribute those low values to low deviatoric stresses, for which there is some micromechanical justification for low n. For ice near glacier beds, however, and particularly near bumps, deviatoric stresses will tend to be high, and thus justification for low values of n is weak.

**MK:** I have rewritten this, and added more evidence from the literature. The reviewer has a point with the low-stress / high stress, and this is now described. However, there is ample evidence that the stress dependency in both experiments and in nature is NOT the 'standard' n=3 that is usually assumed. Note that Chandler's work IS on a real glacier. The main point is that glacial modellers should not assume  n = 3 in temperate ice, if there's no evidence. See revised section on the "Creep Component in Temperate ice"

p. 9, 31. Why does the abbreviation, GBPS, not coincide to the first letters of "grain boundary pressure melting"? And why choose a new term here when there is ample discussion of this sort of deformation in the literature, much of which is not cited here? Overall, I am left with the impression that the author does not have sufficient familiarity with the ice rheology literature.

**MK:** I have rewritten this, with more emphasis on dislocation creep & recrystallisation enhanced by the presence of melt, discussed the potential role of Grain Boundary Sliding, and used the more traditional term 'grain boundary melting' / 'internal pressure melting' (mainly on the instigation of reviewer M Montagnat). There is again much evidence for partial melting in deforming temperate ice, and I've strengthened this. As it is unclear as to which mechanism is dominant (I admit I was pushing too hard for a single mechanism in the original MS), I use the term 'melt-assisted creep' as a bucket term. The problem is that, to my knowledge, no-one has studied the stress-strain behaviour AND the microstructures, which can tell us about the deformation mechanisms. See revised section on the "Creep Component in Temperate ice"

p. 10, 26-27. "In summary, ice flow around a bedrock obstacle in temperate ice is constrained either by stoss-side pressure melting or by enhanced creep." Again this is misleading. Any obstacle is accommodated by both mechanisms, although one can dominate the other depending on bump size.

**MK:** not so much misleading, as poorly phrased. " Ice  flow around a bedrock obstacle in temperate ice is accommodated by stoss-side pressure melting or by enhanced creep, with creep being more important for larger obstacles. In temperate ice, the creep component is….[   }. And the pressure melting is  […. ]".

p. 10, 29-32 & p. 11, 1-4. None of these bullets, or the following assertion, has been demonstrated in this paper.

**MK:** Note:  " it is proposed here that stoss side pressure melting is constrained by ….". Given the preceding matter, these is a very reasonable propositions to make, that are testable. It is clear from the English that I propose a hypothesis here, rather than demonstrate or assert.

p. 10. 5-9. The conclusion here that temperature ice does not obey a power-law rheology has not been demonstrated in this paper.

**MK:** It is now convincingly shown that temperate ice cannot be responsibly modelled according to a standard power-law rheology with the standard n=3. What it should be modelled as maybe unclear, but part of the point of this paper is to point out where and why temperate ice is important, but that there are large gaps in the knowledge of that material. It appears to me that the reviewer thinks that temperate ice behaves the same as cold ice, and can be modelled in the same way. This is clearly not true. Power law cannot be regarded as operating in temperate ice.

p. 11, 19. The idea that no sliding occurs at subfreezing temps is not strictly correct. See Shreve 1984 J Glaciol.; Cuffey et al 1999, GRL.

**MK:** that maybe true, especially near cold/warm boundaries. However, both these papers admit that compared to normal, let alone fast warm-based sliding velocities, the sliding is 'negligible', and therefore as an overall approximation, the scale of which is clear here, it is correct.

Section 8.2. This is an interesting story but no aspect of it has been demonstrated in this paper.

**MK:** The first part is mainstream glaciology, the parts of 3) and 4) follow logically from the paper. A dismissive comment, that is not very useful for an author to act upon. However, I will rephrase the introductory line as: " In such a model, three thermomechanical basal regimes may thus occur, with a potential fourth operating seasonally". Note that reviewer Cohen noted that the figure accompanying this section was very useful.

p. 12, 8-9. "The corollary of the processes described herein is that if a thick temperate layer is present, basal motion over a hard bed with bedrock humps provides less drag than previously thought." This is misleading. The fact that traditional sliding theories, including Weertman's, under-predict rates of glacier sliding and over-predict drag has been discussed for many decades, and is well described in the leading reference book by Cuffey and Paterson (2010), for example. Under-emphasized in this paper is the role of cavity formation in reducing basal drag (e.g., Schoof, 2005).

**MK:** I assume that the reviewer refers to the phrase 'previously thought'; if so, fair point. Rephrased as: "The corollary of the processes described herein is that if a thick temperate layer is present, basal motion over a hard bed with bedrock humps provides less drag than ice modelled with cold ice properties".

p. 12, 11. "weaker bulk rheology"? I can see how ice can be "weak", but I don't understand how the relationship between stress and strain rate (rheology) can be "weak".

**MK:** Rheology is the general study of plastic/viscous deformation, so has a rather wider meaning than merely the stress / strain rate relation – it includes such important things as temperature for instance. In some disciplines, this is then (possibly

erroneously) used in term of 'rheological behaviour' and hence the 'rheology of material X'.    Rephrased as: "Instead they suggest a different, weaker bulk rheological behaviour"

Section 8.3. The problem of fast flow on a hard bed is indeed important and may certainly involve soft ice and the mechanical and thermal effects of water flow under glaciers.

**MK:** exactly, that's part of the point of the paper

The problem is that this paper does not provide new analyses (or data) that convincingly bear on the issue, so these comments on ice streaming come off as speculative and poorly motivated.

**MK:** I stated right in the introduction that this paper is conceptual in character.  It does provide new concepts that do bear on the issue, because, to my knowledge, no-one has invoked a thick temperate layer as an explanation for the NEGIS, nor an explanation for how palaeo-ice streams with poor topographic steering  operated during the Pleistocene. As I understand, the fast flow of the NEGIS is explained either by very high geothermal heat flow that is geologically unreasonable, and for which the presented geophysical evidence (gravity, aeromag) is unconvincing, or by the presence of soft sediment, which is at odds with the rough nature of the bed, as well as with the observations that hard gneiss in deglaciated areas is normally free of till where significant ice sliding has occurred during Pleistocene (see also my Reply to O Eisen).  Thus, there is currently *no* satisfactory explanation for the observed fast ice flow in the NEGIS.  Yes, the existence of a thick temperate layer is speculative, but so are all previous explanations put forward.  Science cannot move forward if hypotheses cannot be proposed: as long as they are testable.  This hypothesis is eminently testable (by drilling or doing radar echo sounding with an appropriate frequency): whether it is correct, time will tell.  I feel it is justified to suggest the possibility of the presence of a thick layer of temperate ice at the base of the NEGIS.  Speculative: yes, but in a responsible way of posing a testable hypothesis; poorly motivated: no, quite the opposite. See also comments by the other reviewers…

[revised manuscript text omitted]

---

## Author Response (AR2)

**Reply to comments.**

**Reviewer 2 comments.**

Major point: The meaning of power-law creep is confusing and this has generated a heated discussion during the review. Probably the confusion arose because of the unclear and undefined meaning of "standard" or "normal" power law or "Glen's flow law". Mathematically the meaning of power-law for ice is clear: strain rate depends on stress to some power. This is what glaciologists usually refer to as Glen's flow law (see Cuffey and Paterson, 4th edition, p. 55). The deformation mechanism, the value of the exponent n in the power-law (whether it is 1.8 or 3 or 4.2), and the Arrhenius function are irrelevant: Glen's flow law is a power-law equation and the value of n is not implicitly fixed when one mentions Glen's flow law (n could be 3.2 as suggested by Glen, 1955, or it could take some other value). If n=1, then it's a linear law. Usually numerical ice flow models use n=3 with some specific values for the prefactor and the activation energy in the Arrhenius relationship and this may be what the author refers to as "standard" or "normal" power law or "Glen's flow law" but the meaning of such qualifier or the definition is not clear in the text (it's clear in the response to reviewer though).

There is no discussion that basal ice at or near the melting temperature behaves differently than cold clean bulk ice. Ice near the melting temperature or ice at low stress may also well have different power-law exponents (closer to 1) and different Arrhenius parameters but they still both behave as power-law materials. If the exponent is exactly one then one can argue that it is not a power-law but a linear material.

Following that strict mathematical definition, Weertman's sliding model (and that of others) is valid for power-law creep regardless of the values of n, the prefactor or the activation energy used. Weertman's specifically chose some particular values for n and A (n=3) but his theory is valid for any combination of n and A.

The data near the melting point in Fig. 5 do not suggest that power-law creep is not dominant (page 7 line 15) because that plot does not show the dependence of strain rate on stress. The exponent of power-law creep could have a different value near the melting temperature depending on the deformation mechanism but the data of Morgan (1991) has no bearing on this as stated by reviewer Montagnat and discussed by the anonymous reviewer. It only indicates a changing Arrhenius parameterization.

To clarify the manuscript, I suggest defining early on the meaning of "standard" power-law as Glen's flow law with n=3 and with parameters for the Arrhenius equation that are commonly used for cold to near-temperate ice (could cite values in Cuffey and Paterson 4th edition). Then the author can argue that this "standard" power-law (used by many including Weertman) is not valid for temperate ice because n=3 does not represent deformation mechanisms operating near the melting temperature and the values of the Arrhenius constants are not appropriate as shown by Morgan 1991. Remove references to Glen's flow law that assume n=3. Call this the "standard" power-law model instead as done almost everywhere in the paper.

In my defence, many review articles (e.g. Weertman 1983) DO take the Arrhenius relation as part and parcel of 'the flow law', and so do other material scientists. Nevertheless, the suggestion of the reviewer is a good one, and I've followed this up: see changes in Section 4 and, where relevant, elsewhere in the manuscript. I've now removed all but two references to 'Glen's flow law.

Minor comments D Cohen:
- Thermal equilibrium: An argument has appeared between the anonymous reviewer and the author because of the use of this term. The author and the reviewer are not using this term in the same sense hence the discord. From a purely thermodynamics point of view, the author is correct, thermal equilibrium means that the temperature is everywhere the same. The reviewer, however, is using a more conventional meaning in

glaciological modeling: equilibrium means steady state conditions, i.e., nothing changes in time but there can spatial variations of the temperature field (or any other fields). In that sense there is equilibrium in Weertman's model but it is not thermodynamic equilibrium. To make the sense clear and avoid confusion because of use of different jargon, may be the first mention of "thermal equilibrium" should indicate that it means a uniform temperature everywhere with no thermal gradient.

Good point. I've rephrased on Page 5 as: *"As a result, the entire basal environment (basal ice, water and top of bedrock) approaches thermal equilibrium near the pressure melting point, with a near-uniform temperature and no significant horizontal thermal gradients."*

Related to that I don't see why ice and brine can't be in thermal equilibrium (page 8 line 31).

Reviewer is entirely correct – this was my mistake, this should have been thermodynamic equilibrium (which includes chemical equilibrium) . Rephrased as: *"This mechanism only operates if ice and water coexist in thermal and chemical equilibrium, and would thus not be observed in experiments with cold ice and where the liquid is a brine. "*

- Write paleo or palaeo but be consistent  MK: Done
- p2, line 19, comma after "warm-based"   MK: Done
- p3 item 2) The question is awkward because it seems to imply that classic sliding models don't assume temperate ice. But they all do.  MK: rephrased.
- p3 line 14 "thermal different controls" -> "different thermal controls" MK: Done
- Fig 1. h_ice = 800 m (small m for meter) MK: Done

- p4 line 9. The work of Emerson and Rempel "the sliding resistance of simulated basal ice", TC, 1, 11-19, 2007, should also be cited.  MK: Done (and changed the order here slightly – more logical).
- p4 Equation 2 would be better rewritten as a conventional melt rate, i.e. with units of length over time (so by dividing everywhere by density). See response to anonymous reviewer   MK: Done
- p5 line 2 Change "Any thermal gradient at" to "Any thermal gradient along" to make clear that this refers to the longitudinal direction, not the vertical. MK: Done
- p6 Fig 5 is cited before Figs 3 and 4. Change figure order.  MK: I've taken out reference to Figure.
- p 11 line 6 Influx is singular so "represents"  MK: Done
- p13 line 3 The last "transport" is not necessary  MK: Done

**Editor 2 comments.**

I have now received two new reviews on the revised version of you manuscript. Both referees have noticed the improvement of this new version and that most of the initial comments were taken into account in the new version. Nevertheless, as mentioned by referee 0000002, the manuscript still need some corrections, especially regarding the definition of what is called "Glen's law" or "Weertman friction law". From my point of view, the manuscript could also be improved by adding a strongest discussion on the form of the friction law that should be used in ice flow models. As a modeler, I found the discussion/conclusion a bit weak in term of suggestion of what should be done to improve current parametrization used in ice flow model. What should be the form of such law? What would be the typical range of the parameters entering such friction law? This would be very helpful for the modeling community. MK:  I've strengthened the discussion a bit and be a bit more specific. However, I'm afraid I can't help you with a clear 'new' sliding law – there's too much uncertainty at the moment to do so!

I have also noticed few minor typos that are listed below.

On the basis of these two new reviews, I think that your paper can be accepted after these minor revisions have been accounted for. From a revised version with changes highlighted as well as a point to point reply to referee 0000002 and my comments, I will take the final decision.

Minor corrections Editor:
- Abstract: is the power-law creep behaviour not applicable for temperate ice is not clear for me. It will be a different exponent and different parameters than the ones used in the classical Glen's law, but it will remain a power-law creep law?  MK: rephrased to 'standard power law. – see comments & reply to Reviewer 2.
-  page 4, line 6: avoid a complete sentence in bracket, or insert it before the point of the first sentence. MK: taken brackets out.
- Figure 1: it should be h_ice = 800 m and not 800 M. In this figure, it is assumed that Q_geo = 50? MK: figure changed; Qgeo comment added.
- page 6, line 25: no space between \epslion and "," MK: done
- page 9, line 7: ice(Tison -> ice (Tison   MK: done
- page 12, line 24: temperature. the rheology -> temperature. The rheology  MK: done
- Appendix: equations that are already written in the main text should not be rewritten here but cited with their number.  MK: done, and re-numbered all equations.
- page 30, line 1: give directly the value of \tau in kPa.  MK: done

[revised manuscript text omitted]